# Water exchange between the Sea of Azov and the Black Sea through the Kerch Strait

Ivan Zavialov[1], Alexander Osadchiev[1], Roman Sedakov[1,2], Bernard Barnier[3,1], Jean-Marc Molines[3], Vladimir Belokopytov[4]

[1]Shirshov Institute of Oceanology, Russian Academy of Sciences, Moscow, Russia.
[2]Moscow Institute of Physics and Technology, Dolgoprudny, Russia.
[3]Institute des Géosciences del'Environment, UGA/CNRS/IRD, Grenoble, France.
[4]Marine Hydrophysical Institute, Russian Academy of Science, Sevastopol, Russia.

*Correspondence to*: Alexander Osadchiev (osadchiev@ocean.ru)

**Abstract.** The Sea of Azov is a small, shallow, and freshened sea that receives large freshwater discharge. Under certain external forcing conditions low-saline waters from the Sea of Azov flow into the northeastern part of the Black Sea through the narrow Kerch Strait and form a surface-advected buoyant plume. Water flow in the Kerch Strait also regularly occurs in the opposite direction, which results in spreading of a bottom-advected plume of saline and dense waters from the Black Sea in the Sea of Azov. In this study we focus on physical mechanisms that govern water exchange through the Kerch Strait, analyse dependence of its direction and intensity on external forcing conditions. Analysis of satellite imagery, wind data, and numerical modelling shows that water exchange in the Kerch Strait is governed by a wind-induced barotropic pressure gradient. Water flow through the shallow and narrow Kerch Strait is a one-way process during the majority of the time. Outflow from the Sea of Azov to the Black Sea is induced by moderate and strong northeasterly winds, while flow into the Sea of Azov from the Black Sea occurs during wind relaxation periods. Direction and intensity of water exchange have wind-governed synoptic and seasonal variability, and do not show dependence on the rate of river discharge to the Sea of Azov on intra-annual time scale. The analysed data reveal dependencies between wind forcing conditions and spatial characteristics of the buoyant plume formed by the outflow from the Sea of Azov.

## 1 Introduction

The Sea of Azov is an enclosed sea located in Eastern Europe and is among the smallest and the shallowest seas in the world (Figure 1). Watershed area of the Sea of Azov (586000 km$^2$) is 15 times greater than the sea area (39000 km$^2$). Therefore, it receives anomalously large river discharge, which annual volume varies in the range of 20 to 54 km$^3$ that is only one order of magnitude smaller than the sea volume (290 km$^3$) (Ross, 1977; Ilyin, 2009). 95% of the annual continental discharge is provided by the Don and Kuban rivers that inflow to the northeastern and southeastern parts of the Sea of Azov, respectively (Ross, 1977; Ilyin, 2009). The southern part of the Sea of Azov is connected to the northeastern part of the Black Sea through the long (45 km) and narrow (4-15 km) Kerch Strait. Hydrological characteristics and the general circulation of the

Sea of Azov are governed by local winds, river runoff, and water exchange with the Black Sea. Low water salinity (1-12) (Ross, 1977; Goptarev et al., 1991; Ilyin, 2009) caused by large freshwater discharge and limited water exchange with the more saline Black Sea (17-18) (Ivanov & Belokopytov, 2011) through the narrow Kerch Strait is one of the main features of the Sea of Azov. Thus, the Sea of Azov is a small, shallow, and brackish water body that can be regarded as the large estuary of the Don and Kuban rivers connected with the Black Sea through the Kerch Strait.

Limited water exchange through a narrow strait hinders mixing between connected water bodies which can result in substantial differences between their physical and chemical characteristics, in particular, concentrations of dissolved and suspended constituents. Thus, transport of water masses through a strait and their subsequent spreading in adjacent sea areas can greatly influence many local processes including coastal circulation, primary productivity, water quality, anthropogenic pollution, and deposition of terrigenous material. Impact of water exchange on these processes depend on, first, physical and chemical characteristics of interacting water masses, and, second, variability of water exchange direction, i.e., frequency, duration, and intensity of water exchange periods.

Many previous studies were focused on physical, biological, and geochemical processes related to water exchange between two large water bodies through a narrow strait in different world regions, in particular, the Baltic and North Seas through the Danish straits (Matthäus and Lass, 1995; Sayin and Kraus, 1996; Jacobsen and Trebuchet, 2000; Sellschoppa, 2006; She et al., 2007), the Black and Mediterranean seas through the Bosphorus and Dardanelles straits (Yuce, 1996; Andersen et al., 1997; Gregg et al., 1999; Falina et al., 2017; Sozer and Ozsoy, 2017; Stanev et al., 2017), the Mediterranean Sea and the Atlantic Ocean through the Strait of Gibraltar (Garret, 1996; Sannino et al., 2002; Beranger et al., 2005, Soto-Navarro et al., 2015), the Bering and Chukchi seas through the Bering Strait (Woodgate et al., 2010, 2012; Danielson et al., 2014), the Patos Lagoon and the Atlantic Ocean (Castelao and Moller, 2006; Marques et al., 2009). A number of papers addressed structure and variability of circulation in the Kerch Strait (Simonov & Altman, 1991; Lomakin et al., 2010, 2016, 2017; Sapozhnikov, 2011; Chepyzhenko, 2015; Kubryakov et al., 2019) and influence of water inflow from the Sea of Azov on the coastal ecosystem in the northeastern part of the Black Sea (Lomakin, 2010; Kolyuchkina et al., 2012; Aleskerova et al., 2017; Izhitsky & Zavialov, 2017; Zavialov et al., 2018). However, many aspects of physical background of water exchange through the Kerch Strait and its dependence on external forcing conditions remain unstudied. Also little attention has been paid to spatial characteristics and temporal variability of sub-mesoscale and mesoscale structures formed in the Black Sea and the Sea of Azov as a result of the water exchange between these seas.

In this study, we address physical mechanisms that drive the water exchange between the Sea of Azov and the Black Sea, using ocean colour satellite imagery, wind reanalysis data, river gauge measurements, and numerical modelling. First, we revealed the dependence of direction and intensity of water exchange through the Kerch Strait on external forcing conditions. Second, we analyzed dynamics of the surface-advected plume of brackish waters from the Sea of Azov spreading in the Black Sea, hereafter, referred as AP, and bottom-advected plume of saline waters from the Black Sea spreading in the Sea of Azov, hereafter, referred as BP. Finally, we revealed dependence of spatial characteristics of AP on external forcing conditions.

The paper is organized as follows. Section 2 provides the detailed information about the study region. Satellite, wind, and river discharge data, as well as the numerical model used for simulation of sea circulation in this study region are described in Section 3. Section 4 focuses on the dynamics of inflow and spreading of AP in the Black Sea and BP in the Sea of Azov and addresses dependence of these processes on the external forcing conditions on synoptic time scale. The discussion of the obtained results followed by the conclusions is given in section 5.

## 2 Study Area

The Sea of Azov is small and shallow, its average and maximal depths are 7 and 14.4 m. The central part of the Sea of Azov is 10-13 m deep and accounts for less than 50% of the sea area (Figure 1). The southern part of the Sea of Azov is connected with the northeastern part of the Black Sea by the Kerch Strait. The narrowest passages of the Kerch Strait are located at its northern (4-5 km) and central (3 km) parts, while at its southern part strait width increases up to 15 km. The central part of the Kerch Strait is very shallow (3-5 m) and steadily deepens to depths of 10 and 20 m at its northern and southern parts, respectively. Bathymetry of the northeastern part of the Black Sea is characterized by the narrow shelf, the distance from the shore to the 100 m isobath varies between 15 and 30 km. (Figure 1).

Large freshwater discharge strongly influences the Sea of Azov. The Don River is the largest river inflowing to the Sea of Azov providing approximately 65% of total freshwater runoff to the sea, the Kuban River provides another 30% of total freshwater runoff. Volumes of annual discharge of the Don and Kuban rivers vary from 18 to 28 $km^3$ and from 6 to 13 $km^3$, respectively, which is caused by strong climatic and anthropogenic influence (Goptarev et al., 1991; Ilyin et al., 2009). Flow regimes of the Don and Kuban rivers have long spring-summer freshet during March – June and April – July, respectively. However, discharges during these periods are only twice larger than during the rest of the year (Goptarev et al., 1991; Ilyin et al., 2009). Difference between evaporation and precipitation over the Sea of Azov (17 $km^3$) is half less than the mean annual river runoff (35 $km^3$) and shows very low inter-annual variability (Ilyin et al., 2009).

Surface temperature of the Sea of Azov is prone to large seasonal variability from 0 °C in winter to 25 °C in summer (Goptarev et al., 1991; Ilyin et al., 2009). Sea ice covers the northern part of the Sea of Azov every year from December – January to March – April, while its central and southern parts are frozen only during extremely cold winters, which occurred only twice during the last 40 years (Ilyin et al., 2009). Surface salinity in the Sea of Azov varies from 9 to 13 except the most freshened northeastern part, namely, the Taganrog Bay, which receives discharge of the Don River. Wind-induced mixing penetrates to sea bottom which results in low gradients in vertical thermohaline structure of the Sea of Azov (Goptarev et al., 1991; Ilyin et al., 2009). Temperature in the surface layer of the northeastern part of the Black Sea also varies largely from 7 °C in winter to 23 °C in summer, however, its salinity is stable (17-19) during the whole year (Ivanov & Belokopytov, 2011). Large freshwater discharge and intense wind-induced vertical mixing result in high concentrations of terrigenous sediments, nutrients, and chlorophyll a in the Sea of Azov, which is one order of magnitude greater than in the northeastern part of the Black Sea (Ilyin et al., 2009).

Circulation in the Sea of Azov is mainly wind-driven, while baroclinic forcing is weak (Cherkesov & Shul'ga, 2018). As a result, sea current field and level of the Sea of Azov are prone to large synoptic variability caused by intense wind surges, which amplitudes regularly exceed 2 m (Ivanov, 2011; Fomin, 2015; 2017). Circulation in the surface layer in the northeastern part of the Black Sea is dominated by, first, the westward current along the continental slope (0.2–0.5 m s$^{-1}$),

which is a part of the Black Sea Rim Current, and, second, the anticyclonic eddy, which is regularly formed between this current and the coast near the Kerch Strait (0.05–0.4 m s$^{-1}$) (Oguz et al., 1993; Ginzburg et al., 2002; Zatsepin et al., 2003; Korotaev et al., 2003). Tidal amplitudes at the northeastern part of the Black Sea and in the Sea of Azov are 2-4 cm; thus, tidal circulation is very low at the study area (Medvedev et al., 2016; Medvedev, 2018).

Water transport through the Kerch Strait is an important part of the water budget of the Sea of Azov, however, its

characteristics are prone to large uncertainty. Volumes of annual water inflow from the Sea of Azov to the Black Sea and in the opposite direction are estimated as 35-64 and 26-44 km$^3$, respectively Current velocities in the Kerch Strait generally exceed 10 cm s$^{-1}$, mean current velocities in its narrowest part are 20-30 cm s$^{-1}$. Barotropic tidal current in the Kerch Strait is less than 5 cm s$^{-1}$ except the narrowest part of the strait where they are equal to 6-10 cm s$^{-1}$ during the peak flow (Ferrain et al., 2018). However, maximal velocity of the 2-days averaged tidal current in the Kerch Strait is less than 5 cm s$^{-1}$ and its

flow direction reverses during the tidal cycle. Thus, tidal currents do not form persistent residual flow and their role in water exchange between the Black and Azov seas can be regarded as negligible.

Large salinity difference between the Azov and Black seas results in substantially different spreading and mixing dynamics of waters that inflow from the Sea of Azov to the Black Sea, on the one hand, and waters that inflow from the Black Sea to the Sea of Azov, on the other hand. Inflow of brackish waters from the Sea of Azov to the Black Sea forms a surface-

advected AP, which is spreading over wide areas (up to 2000 km$^3$) in the northeastern part of the Black Sea (Aleskerova et al., 2017; Kubryakov et al., 2019). Due to elevated concentrations of terrigenous sediments, nutrients, and anthropogenic pollutants in the waters of the Sea of Azov, AP strongly influences physical, biological, and geochemical processes in the areas adjacent to the Kerch Strait in the northeastern part of the Black Sea (Lomakin, 2010; Kolyuchkina et al., 2012; Aleskerova et al., 2017; Izhitsky and Zavialov, 2017; Zavialov et al., 2018). AP is regularly entrained by mesoscale eddies

formed in the Black Sea near the Kerch Strait that can significantly intensify cross-shelf transport of low-saline water in the Black Sea (Kubryakov et al., 2019).

Processes of inflow, spreading, and mixing of Black Sea water in the Sea of Azov received much less attention. Saline waters from the Black Sea form a bottom-advected BP, which can affect large areas in the Sea of Azov. However, its characteristics, spatial structure, and temporal variability remain mainly unstudied.

# 3 Data and methods

## 3.1 Data used

Satellite data used in this study include satellite imagery from EnviSat MERIS with a spatial resolution of 300 m provided by the European Space Agency (ESA) and from Terra MODIS and Aqua MODIS with a spatial resolution of 250 m provided by the National Aeronautics and Space Administration (NASA). MERIS L1 satellite products were downloaded from the ESA web repository (http://merisfrs-merci-ds.eo.esa.int/merci) and used for retrieving maps of sea surface distributions of total suspended matter (TSM) and chlorophyll (Chl-a) using the MERIS Case-2 Regional water processing module (Doerffer and Schiller, 2008). MODIS L1 satellite products were downloaded from the NASA web repository (https://ladsweb.modaps.eosdis.nasa.gov/) and used for retrieving maps of sea surface distributions of TSM and Chl-a using the MSL12 processing module. We analyzed 152 MERIS and 155 MODIS satellite images of the study region taken in 2002–2012 and 2012–2019, respectively.

The Don and Kuban discharge and local wind forcing data were used to study influence of external forcing conditions on the water exchange through the Kerch Strait and spreading of AP in the northeastern part of the Black Sea. The Don and Kuban daily discharge data were obtained from the Razdorskaya and Temryuk gauge stations, respectively, while local wind measurements were performed at the Kerch meteorological station (Figure 1). The atmospheric influence was also examined using wind data obtained from a 6 h NCEP/NCAR reanalysis with a 2.5-degree resolution which showed good accordance with in situ data for the study region (Garmashov et al., 2016). We used zonal and meridional wind components from the only reanalysis grid point located at the study area (44.7611° N, 35.625° E) which were validated against the in situ wind measurements.

## 3.2 Identification of AP and BP by satellite imagery

As it was discussed in Section 3.1, waters of the Azov and Black seas have very different physical and chemical properties. As a result, various ocean surface characteristics measured by satellite instruments can be used to study spreading of surface-advected AP in the northeastern part of the Black Sea. Previous related studies used sea surface temperature (SST), concentrations of Chl-a and TSM retrieved from optical satellite data (Ivanov & Belokopytov, 2011; Aleskerova et al., 2017; Kubryakov, 2019). However, all these characteristics cannot be used for straightforward identification of inflow of brackish waters from the Sea of Azov to the Black Sea, hereafter referred as AI, and identification of boundaries of AP due to the following reasons.

First, difference in SST between the southern part of the Sea of Azov and the northeastern part of the Black Sea varies from –4 °C in winter to +4 °C in summer (Aleskerova et al., 2017). On the other hand, the diurnal variability of SST in the coastal areas of the Sea of Azov and the Black Sea is also equal to several degrees (Ivanov & Belokopytov, 2011; Chepyzhenko, 2015). This fact can prevent formation of distinctive frontal zones between AP and the adjacent sea, especially if AP is formed by inflow from the Sea of Azov during several days, i.e. several diurnal temperature cycles of SST. Mean seasonal

and diurnal variability of TSM is much lower than the difference between the mean values of TSM in the Azov and Black seas during the whole year (Lomakin, 2017). Settling time of fine fraction of suspended sediments within the AP is much lower than residence time of AP in the Black Sea, i.e., AP mixes with ambient saline water and dissipates more quickly than fine suspended sediments settle from surface layer to subjacent sea. Thus, TSM provides a clear optical signal of turbid AP

in the Black Sea and forms stable gradients at borders of AP, which are distinctly visible at optical satellite imagery. However, wind-induced resuspension of sea bottom sediments, which regularly occurs along the northeastern coast of the Black Sea, causes increase of TSM, which can exceed mean TSM values of AP (Figure 2a). Chl-a, on the opposite to TSM, is characterized by, first, larger seasonal variability of its difference between the Azov and Black seas during the whole year and, second, larger synoptic variability within the AP caused by complex biological processes. On the other hand, Chl-a has

lower short-term variability, in particular, in response to wind forcing.

As it was shown above, surface distributions of SST, TSM, and Chl-a in the study region are prone to substantial variability defined by various processes apart from mixing between waters from the Black and Azov seas. However, these processes and their temporal scales are different for SST (diurnal cycle of solar radiation), TSM (episodic wind-induced bottom resuspension events), and Chl-a (synoptic and seasonal biological cycles). Thus, joint analysis of SST, TSM, and Chl-a

distributions can be used for accurate detection of spreading of AP in the Black Sea. We applied the following scheme of identification of AI events and detection of borders of AP based on satellite data. Inflow events were identified by elevated concentration of Chl-a in the Kerch Strait and the adjacent coastal area of the Black Sea, because Chl-a has the lowest short-term variability among the considered sea surface characteristics. If an inflow event was detected, we analyzed areas of elevated TSM, Chl-a, and elevated (in summer) or reduced (in winter) SST associated with formation of AP in the

northeastern part of the Black Sea. If general forms and spatial scales of these areas were similar, we defined borders of AP based on gradient of TSM, that is the most stable passive tracer of AP in absence of episodic wind-induced bottom resuspension events (Figure 2b). If areas of TSM, Chl-a, and SST anomalies were not consistent with each other, we assumed that areas of elevated TSM and reduced SST (in winter) were modified by wind-induced resuspension and mixing of AP with subjacent sea. In this case we defined borders of AP based on gradient of Chl-a, that is the most stable tracer of

AP under intense wind forcing conditions during the whole year.

Inflow of saline waters from the Black Sea to the Sea of Azov, hereafter referred as BI, causes formation of the bottom-advected BP that cannot be directly identified at satellite imagery. However, wind-induced mixing, which regularly penetrates to sea bottom at the shallow Sea of Azov, can cause mixing of water of BP with overlaying water of the Sea of Azov. It results in reduced values of TSM and Chl-a in the surface layer above the spreading area of BP, as compared to the

adjacent areas of the Sea of Azov. Thus, presence of BP in the bottom layer can be identified at satellite imagery as areas of reduced TSM and Chl-a in the Sea of Azov adjacent to the Kerch Strait. The scheme of identification of BI events and detection of borders of BP at satellite imagery is relatively straightforward, as compared to identification of AI events. However, we assume that BP is not manifested by anomalies of TSM and Chl-a in the surface layer during low wind forcing conditions, while during strong wind forcing conditions resuspension of bottom sediments can induce elevated

concentrations of TSM in the surface layer that also hinders identification of BP. As a result, many of BI events are not detected by optical satellite imagery.

### 3.3 Numerical model

In this study we performed numerical simulations using the BSAS12 numerical model to simulate circulation in the Black and Azov seas and study the water exchange through the Kerch Strait. BSAS12 is an original regional configuration of the ocean and sea-ice general circulation model NEMO (version 3.6) that covers the Black and Azov seas (Madec, 2016). Horizontal grid resolution of the model is 1/12° that is approximately 6.75 km in the study region. The vertical coordinate is represented by 59 vertical z-levels with the finest resolution (1 m) at the upper ocean. A partial-step representation of bottom topography which adjusts vertical size of the model bottom level to the real ocean depth is used (Barnier et al., 2006). The model domain has an open ocean boundary at the Bosporus Strait that connects the Black Sea with the Mediterranean Sea. The ocean is driven by the ERA-Interim atmospheric forcing which includes 3-hourly fields of near surface wind velocity, temperature, and humidity and daily fields for incoming long and short wave radiation and total precipitation. Surface fluxes and wind stress are calculated using the CORE bulk formulae (Large and Yeager, 2004) using sea surface temperature provided by the model. The initial temperature and salinity fields of the Azov and Black seas are obtained from the climatological data given in Goptarev et al. (1991) and Belokopytov (2018). BSAS12 is forced by monthly climatological river runoff to the Black and Azov seas that was set according to the data provided in Jaoshvili (2002), Dai and Trenberth (2002), and gauge data from the Don and Kuban rivers described in Section 3.1. The boundary conditions at the Bosporus Strait are prescribed according to the data provided by Gregg et al. (1999, 2002), Altiok et al. (2012), and Sozer and Ozsoy (2017). The model time step is set to 720 s. The BSAS12 simulates circulation of the Black and Azov seas during the period from 1 January 1992 to 31 December 2017. In this study we focus on modelling of circulation in the areas of the Black and Azov seas adjacent to the Kerch Strait with emphasis on the water exchange through the strait. The main large-scale and mesoscale circulation features of the Black Sea circulation were adequately reproduced by the numerical modeling (Figure 3), including the Black Sea Rim Current, the quasi-stationary cyclonic gyres at the central divergence zone, and multiple quasi-stationary anticyclonic gyres between the Rim Current and the shoreline near Sebastopol, Batumi, etc. (Oguz et al., 1992; 1993; 1995; Stanev, 1995; Staneva et al., 2001). The model also reproduced well seasonal variations of sea surface circulation, in particular, winter–spring intensification of the Rim Current, meandering of the main flow of the Rim Current caused by baroclinic instability, and formation of multiple nearshore anticyclonic eddies during summer at the eastern part of the Black Sea (Oguz et al., 1992; 1993; Titov, 2002; Zatsepin et al., 2003; Enriquez et al., 2005). The mean annual values of the water transport through the Kerch Strait during the modelling period (Figure 4) show good agreement with the reference values of 20 km$^3$ (Stanev, 1990). A short assessment of the surface circulation produced by the model is presented in a Supplementary Material.

## 4 Results

### 4.1 Water exchange through the Kerch Strait

We used the BSAS12 model to study physical mechanisms that govern the water exchange through the Kerch Strait. Based on the simulation outputs, we reconstructed daily averaged baroclinic and barotropic components of pressure gradient force in the Kerch Strait during 1992 – 2017. Correlation analysis shows that the total pressure gradient along the strait is mostly governed by the barotropic component (R = 0.7), while the role of the baroclinic component is smaller (R = 0.3). This feature is caused by relatively large average difference of water level (< 0.1 m during the majority of a year) in the southern and northern ends of the strait. Local wind forcing induces large synoptic variability in magnitude and direction of the barotropic pressure gradient. Stable density jump that exists along the strait does not exceed 6 kg/m$^3$, therefore, the baroclinic pressure gradient is one order of magnitude smaller than the barotropic pressure gradient, and does not induce a steady exchange circulation typical for positive estuaries. As a result, circulation through the Kerch Strait is not steady and unidirectional, but has large synoptic variability of intensity and direction governed by episodic wind forcing events. Annual variability of the total pressure gradient in the strait does not show any seasonality. Thus, water exchange through the Kerch Strait has wind-govern synoptic variability, and does not show seasonal dependence on river discharge rate to the Sea of Azov.

The role of the barotropic component in the total pressure gradient is the largest in the most shallow (3-4 m) and narrow (3 km) parts of the Kerch Strait. Numerical simulations revealed that even moderate wind forcing at the study region induces a one-way water transport, i.e., only inflow or only outflow, in this part of the strait, which defines the water exchange between the Azov and Black seas. Water flow occurs simultaneously in both directions in the shallow part of the strait, i.e., a two-way water transport is formed, only during light winds. Numerical simulations show that a two-way water exchange occurred in the Kerch Strait only during weak wind forcing conditions, which total annual duration was 34 – 54 days in 1992 – 2010, that is only 9-15% of a year (Table 1). Thus, a one-way water exchange between the Azov and Black seas was observed during the majority of the year, which is not typical for a positive estuary. This result is also supported by previous studies based on in situ observations (Ivanov, 2011) and numerical modelling (Stanev et al., 2017) of water exchange in the Kerch Strait.

### 4.2 Spreading of AP in the Black Sea

River discharge to the Sea of Azov and wind forcing at the area of the Kerch Strait are the main factors that are believed to govern AI events (Goptarev et al., 1991; Simonov & Altman, 1991; Ivanov & Belokopytov, 2011). We analysed 68 AI events identified at MERIS and MODIS optical satellite imagery during 2002-2019 and verified them by the BSAS12 simulations. Based on wind reanalysis data and gauge data of the Don and Kuban rivers, we studied dependence of formation of AI events on wind forcing conditions and river discharge on synoptic and seasonal time scales.

First, we analysed the relation between concentration of Chl-a in the Kerch Strait and the adjacent area of the Black Sea retrieved from optical satellite imagery, on the one hand, and direction of wind forcing averaged over 24 hours preceding satellite observations, on the other hand (Figure 5). Elevated concentrations of Chl-a, which is regarded as the main indicator of an AI event, were observed only if azimuthal angle of wind direction was between 30º and 80º and wind velocity

exceeded 5 m/s. Thus, formation of an AI event is induced only by moderate and strong northeasterly winds. Second, we addressed the relation between concentration of Chl-a and discharge rates of the Don and Kuban Rivers (Figure 6). We obtained that AI events are formed under the whole variety of discharge conditions and do not show any dependence on discharge rate on synoptic time scale. Synoptic variability of river discharge strongly influences water exchange between a river estuary and open sea if volume of an estuary is relatively small as compared to river discharge rate (Officer, 1976;

Sheldon & Alber, 2002; Wang et al., 2004). However, the volume of the Sea of Azov, regarded as the estuary of the Don and Kuban rivers, is one order of magnitude greater than the annual freshwater runoff. As a result, signal of synoptic variability of river discharge dissipates in the Sea of Azov and does not influence formation of AI events. Thus, we obtain that AI events are induced by wind forcing and do not depend on river discharge conditions on synoptic time scale.

Spreading of a buoyant plume in a non-tidal sea is mainly governed by discharge rate and wind forcing (Fong et al., 1997;

Hallock & Marmorino, 2002; Horner-Devine et al., 2015; Osadchiev and Sedakov, 2019) and is characterized by strong variability of size, shape, and spreading patterns under different configurations of external forcing conditions (Kourafalou et al., 1996; Xia et al., 2007; Zavialov et al., 2014; Osadchiev et al., 2016; 2017; Osadchiev and Korshenko, 2017). However, analysis of satellite imagery and numerical modelling revealed stability of the spreading pattern of AP. All satellite images where AI event was detected showed that AP propagated westward along the southeastern shore of the Crimean Peninsula

(Figure 2b, Figure 7). This result is supported by the BSAS12 numerical simulations that showed good agreement between modelled distribution of low saline AP and location of AP detected at optical satellite imagery. This freshened alongshore current formed by AP dissipated on a distance of 50-200 km from its source at the Kerch Strait. AP did not spread eastward along the coast of the Taman Peninsula or southward to the open sea. Elevated values of TSM to the east from the Kerch Strait along the Taman Peninsula, which are regularly observed at satellite imagery, are neither accompanied by elevated

values of Chl-a nor low surface salinity (Figure 2a). This fact indicates that these turbidity features are induced by bottom resuspension and do not correspond to eastward spreading of AP along the coast of the Taman Peninsula.

Stability of the spreading pattern of AP can be explained in the following way. As it was shown above, AI occurs only during northeasterly wind and causes formation of AP. Thus, initial spreading of AP from its source at the Kerch Strait is forced by northeasterly wind. As a result, the AP forms a quasi-geostrophic coastal current in response to downwelling-

favourable wind forcing, which was addressed in many previous studies (Garvine, 1987; Yankovsky & Chapman, 1997; Fong and Geyer, 2002). Spreading of AP in different direction, e.g., southward or eastward, requires change of direction of wind forcing. However, change of wind direction results in cessation of inflow of brackish water to the Black Sea and causes dissipation of the freshened alongshore current.

Satellite imagery and numerical simulations show that AP occupy a wide area along the southeastern shore of the Crimean Peninsula in case of stable inflow of brackish waters from the Sea of Azov to the Black Sea during 3-5 days (Figure 7). Alongshore extent and area of AP can increase by one order of magnitude and exceed 150 km and 2000 km$^2$, respectively, during an individual AI event. After secession of inflow from the Sea of Azov to the Black Sea, AP dissipates during several

days. As it was shown above, intensity of AI, i.e., freshwater discharge rate, depends on local wind, which is, therefore, the only critical external force that governs synoptic variability of spatial scales of AP. Thus, we can reconstruct dependence of spatial characteristics of AP identified at satellite imagery on speed and duration of northeasterly wind forcing. For this purpose, we used the wind forcing index $W_t = \tau_t \cdot t$, where $\tau_t$ is the average wind stress during the time period $t$ when wind direction was between 30 and 80º. Alongshore extent and area of AP were identified at 68 satellite images obtained during or

shortly after northeasterly wind forcing conditions. For every registered AI event we calculated values of alongshore extent ($L$) and area ($S$) of AP and compared them with values of wind forcing index $W_t$ for the related periods of predominant northeasterly wind forcing preceding satellite observations. For all these cases $W_t$ exceeded 332.8 s·N/m$^2$, therefore, we presume this value as the threshold for formation of an AI event. Alongshore extent and area of AP increase with increase of wind forcing index, however, this increase is not steady. In particular, alongshore extent and area of AP are almost stable if

the wind forcing index exceeds $7 \cdot 10^4$ s·N/m$^2$. Thus, the observed forms of dependences between have good approximation by logarithmic functions. Figure 8 illustrates the obtained relations between $L$ and $S$, on the one hand, and $W_t$, on the other hand: $L = -186.9 + 74.1 \cdot \lg(W_t)$ (RMSE is about 15 km); $S = -2537.7 + 838.6 \cdot \lg(W_t)$ (RMSE is about 163 km$^2$).

In this study we did not consider the impact of ambient sea currents of the Black Sea on spreading of AP along the Crimean Peninsula. This impact is not negligible, in particular, intensification of the Rim Current enhances along-shore spreading of

AP, while the mesoscale anticyclonic eddy formed between the Rim Current and the Kerch Peninsula induces cross-shore spreading of AP (Kubryakov, 2019). However, the obtained relations and numerical modelling results show that spreading of AP is mainly governed by wind forcing, while coastal circulation plays a secondary role which is typical for buoyant surface-advected water masses (Washburn et al., 2003; Ostrander et al., 2008; Osadchiev and Zavialov, 2013; Osadchiev, 2015).

Finally, we analysed dependence of seasonal variability of area of AP, which is indicative of intensity of AI, on wind forcing and river discharge conditions. For this purpose we calculated monthly averages of area of AP detected at optical satellite imagery and compared them with monthly averages of the wind forcing index and total discharge rate of the Don and Kuban rivers (Figure 9). Monthly variability of average area of AP shows the direct relation with monthly variability of the wind forcing index. The obtained graph reveals that both characteristics have similar monthly variations with two distinct peaks in

September and December – March, while the lowest values are registered in May – June. However, the opposite situation is observed for dependence between monthly averages of area of AP and river discharge rate. Several studies showed that variability of the Don and Kuban discharges induces variability of level of the Sea of Azov, thus, presumably influencing formation of AI events on seasonal time scale (Goptarev et al., 1991; Filippov, 2015). Nevertheless, seasonal variability of

river discharge characterized by distinct spring freshet and autumn-winter draught showed no relation with intensity of AI events. Thus, we obtain that river discharge does not significantly affect seasonal variability of AI.

## 4.3 Spreading of BP in the Sea of Azov

Inflow of saline waters from the Black Sea to the Sea of Azov causes formation of dense BP that it is spreading in the bottom layer and cannot be directly identified at satellite imagery. As it was described in Section 2.2, BP can be identified at satellite imagery as area of reduced TSM and Chl-a in the Sea of Azov adjacent to the Kerch Strait (Figure 10). We analysed MERIS and MODIS optical satellite imagery acquired during 2002-2019 and identified only 8 BI events which were confirmed by salinity distributions obtained from the BSAS12 numerical simulations. Thus, we assume that BP is not manifested by anomalies of TSM and Chl-a in the surface layer during low wind forcing conditions and many of BI events are not detected at optical satellite imagery.

All identified BI events were preceded by strong north-easterly winds, which caused intense outflow from the Sea of Azov to the Black Sea. Numerical simulations showed that formation of BI events was caused by reverse of the barotropic pressure gradient along the Kerch Strait as a result of relaxation of strong northeasterly wind forcing. Figure 10 illustrates typical cases of formation of BI events in response to local wind forcing. Strong northeasterly wind (up to 15 m/s) observed during 14-21 April 2003 caused formation of AP, which was followed by light and variable wind on 21-27 April 2003 and formation of BP detected at satellite imagery on 27 April 2003 (Figure 10a). Formation of BP detected at satellite imagery on 12 November 2015 was also preceded by, first, strong northeasterly wind (up to 10 m/s) during 29 October – 1 November 2015 that caused formation of AP and, second, light and variable wind on 2-12 November 2015 (Figure 10b).

Similar configuration of the water exchange between the large Patos Lagoon and the Atlantic Ocean through the narrow strait was described by Castelao and Moller (2006). They revealed formation of reverse flow of saline ocean water to the lagoon after outflow of brackish lagoon water induced by wind forcing. Our results are also supported by the in situ measurements of vertical current profiles performed reported by Ivanov (2012). These measurements showed that moderate northeasterly winds observed on 26-27 September 2011 induced surface-to-bottom southward flow in the shallow northern part of the Kerch Strait. Light westerly wind observed on 28 September 2011 resulted in termination of the southward surface flow and formation of the distinct surface-to-bottom northward flow.

## 5 Discussion and Conclusions

In this work we studied the water exchange between the Sea of Azov to the Black Sea through the Kerch Strait. We revealed that different physical mechanisms govern the water transport in southward (from the Sea of Azov to the Black Sea, AI events) and northward (from the Black Sea to the Sea of Azov, BI events) directions. Analysis of satellite imagery, wind data, and numerical model outputs shows that the water exchange in the Kerch Strait is governed by the wind-induced barotropic pressure gradient. As a result, the water flow through the shallow and narrow Kerch Strait is a one-way process

during the majority of the time. Southward AI events are induced by moderate and strong northeasterly wind forcing. In this case, wind stress causes transport of brackish water from the shallow southern part of the Sea of Azov through the Kerch Strait, which results in formation of the buoyant plume in the Black Sea. This surface-advected Azov Sea water plume (AP) is characterized by elevated concentrations of suspended sediments and chlorophyll a and can be detected at optical satellite

imagery. AP is spreading off the Kerch Strait as a quasi-geostrophic coastal current along the southeastern shore of the Crimean Peninsula, its area steadily increases during an AI event that last 1-5 days. As a result, AP occupies a large area in the northeastern part of the Black Sea up to 2000 $km^2$. However, AP dissipates during 1-5 days after end of AI event. The short-term, but regular process of formation and spreading of AP at the northeastern part of the Black Sea influences many local physical, biochemical, and geological processes, which were addressed in many previous studies.

The northward water transport in the Kerch Strait, on the opposite, was registered after relaxation of strong northeasterly winds that results in reverse of the barotropic pressure gradient along the strait and triggers inflow from the Black Sea to the Sea of Azov. Thus, strong northeasterly wind plays a restricting role for this process, because the intense wind-induced southward surface flow of water from the Sea of Azov occupies the whole water column in the shallow northern part of the Kerch Strait. Analysis of satellite images did not show any direct dependence of the northward water transport in the Kerch

Strait on characteristics of local wind forcing. However, this feature can be caused by relatively low number of detected BI events at satellite imagery. Future studies of the role of wind forcing in this process require more specific and detailed in situ measurements and/or numerical modelling.

We determined that wind forcing governs direction and intensity of water transport in the Kerch Strait on inter-annual time scale. River runoff to the Sea of Azov does not show any distinct influence on synoptic and seasonal variability of the water

exchange through the Kerch Strait, i.e., the signal of river discharge dissipates in the Sea of Azov and does not influence freshwater outflow from the estuary to the open sea. This feature is not typical for large river estuaries, e.g., the Patos Lagoon (Castelao and Moller, 2006) and the Amur Liman (Osadchiev, 2017). The relation between freshwater discharge to the river estuary and the water exchange between the estuary and the open sea depends on two main factors, namely, the volume of inflowing river discharge and the spatial scales of an estuary. The volume of the Sea of Azov (290 $km^3$) is one

order of magnitude greater than the annual continental discharge to the sea (20 – 54 $km^3$). River runoff during flooding periods increases the level of the Sea of Azov only by 6-7 centimetres as compared to draught periods. As a result, signal of seasonal discharge variability of the Don and Kuban rivers dissipates in the Sea of Azov and does not influence the intensity of the water exchange through the Kerch Strait. In contrast, the volume of the Patos Lagoon (50 $km^3$) is of the same order as continental runoff volume (75 $km^3$), which causes an increase of the lagoon level by 70-80 cm during wet periods. As a

result, the stable seaward flow from the Patos Lagoon during the seasonal flood can be reversed only by very strong winds (Moller and Castaing, 1999). The volume of the Amur Liman (20 $km^3$) is much less than the annual Amur discharge volume (400 $km^3$). Therefore, the water exchange between the Amur Liman and the open sea is dominated by the river regime during the major part of a year (Osadchiev, 2017).

Based on the quantitative relations between the spatial characteristics of AP and local wind forcing established in Section 4.2 we can reconstruct dependence of discharge rate from the Sea of Azov to the Black Sea through the Kerch Strait based on speed and duration of local wind forcing. First, we define "AI-favorable" wind conditions as a period of predominant northeasterly wind forcing with the wind-forcing index $W_t$ exceeding 332.8 s·N/m$^2$. Second, using wind reanalysis data we identified periods of AI-favourable wind events. Finally, we calculated areas of AP formed during these events using the obtained relation between wind forcing index and area of AP. The sum of these areas for all AI events occurred during a year ($S_{AP}$) is indicative of the total annual volume of water inflow from the Sea of Azov to the Black Sea through the Kerch Strait ($V_{AP}$). Simonov & Altman (1991) estimate mean value of $V_{AP}$ during 1963-1972 as 64.3 km$^3$, while we calculated that mean value of $S_{AP}$ during 1963-1972 is equal to 16571 km$^2$. We assume that the mean depth of AP ($H_{AP}$) does not substantially depend on its spatial scale and is mainly defined by bathymetry of the Kerch Strait. Thus, we obtain that $V_{AP} = H_{AP} S_{AP}$ and the mean annual depth of AP is $H_{AP} = V_{AP} / S_{AP} = 4$ m, which is consistent with depth of the Kerch Strait (Figure 1). Finally, using the dependence of the plume area on the wind forcing index and the obtained estimate of the mean depth of AP we set the following equation for the water inflow volume during an AI event ($V$): $V = S \cdot H_{AP} = -10150.8 + 3354.4 \cdot \lg(W_t)$.

Many numerical studies were focused on circulation, food webs, water quality, transport and fate of dissolved and suspended matter, and other processes in the Black Sea (e.g., Stanev, 1990; Oguz et al., 1995; Stanev & Staneva, 2000; Staneva et al., 2001; Enriquez et al., 2005; Korotenko et al., 2010; Korotenko, 2017, Stanev et al., 2017). of the majority of these studies did not simulate circulation in the Sea of Azov and the Kerch Strait, but reproduced the water exchange through the Kerch Strait as a boundary condition. However, these works generally applied mean annual or mean seasonal exchange values and neglected the fact that direction and discharge rates of the water transport through the Kerch Strait have strongly inhomogeneous temporal distributions and significant inter-annual variabilities. In particular, we are not aware of any relevant numerical parameterizations of water exchange through the Kerch Strait that reproduce its synoptic variability. Thus, the equations, which define conditions of formation of AI events and dependence of discharge rate during AI event on speed and duration of northeasterly wind, that were obtained in the current study hold promise to be useful for numerical modelling of processes in the Black Sea. They can improve the existing parameterizations of the boundary conditions at the Kerch Strait and, therefore, increase accuracy of numerical simulations of physical, geological, and biochemical processes in the Black Sea.

## Data availability

The Envisat MERIS satellite data were downloaded from the European Space Agency repository of the Envisat satellite data http://merisfrs-merci-ds.eo.esa.int/merci (available after registration). The Terra MODIS and Aqua MODIS satellite data were downloaded from the National Aeronautics and Space Administration repository of LANCE-MODIS satellite data https://lance3.modaps.eosdis.nasa.gov. The river discharge and wind data were downloaded from the Federal Service for Hydrometeorology and Environmental Monitoring of Russia repositories http://gis.vodinfo.ru (available after registration)

and https://rp5.ru. The NCEP/NCAR reanalysis data were downloaded from the National Oceanic and Atmospheric Administration website https://www.esrl.noaa.gov/psd/data/gridded/data.ncep.reanalysis.surface.html.

## Competing interests

The authors declare that they have no conflict of interest.

## Acknowledgements

This research was funded by the Ministry of Science and Education of Russia, theme 0149-2019-0003, theme 0149-2019-0003 (collecting and processing of wind, river discharge, and satellite data), the Russian Foundation for Basic Research, research project 18-05-80049 (study of water exchange in the Kerch Strait), the Russian Science Foundation, research project 18-17-00156 (study of spreading of the Azov plume in the Black Sea), and the Russian Ministry of Science and
Higher Education, research project 14.W03.31.0006 (developing of numerical modelling). Contribution to the BSAS12 modelling effort also came from the joint French-Russian cooperation program PHC Kolmogorov No 38102RF (conducting of numerical experiments). The computations presented in this study were performed at the Centre Informatique National de l'Enseignement Supérieur (CINES) under the allocation made by GENCI A0050100727.

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

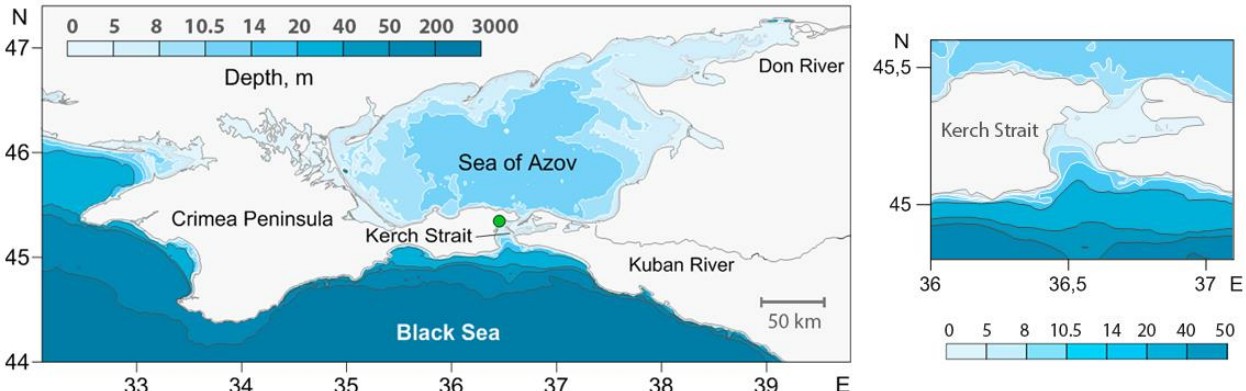

**Figure 1: Bathymetry of the Sea of Azov and the northeastern part of the Black Sea (left) and the Kerch Strait (right). Locations of the estuaries of the Don and Kuban rivers and the Kerch meteorological station (green circle).**

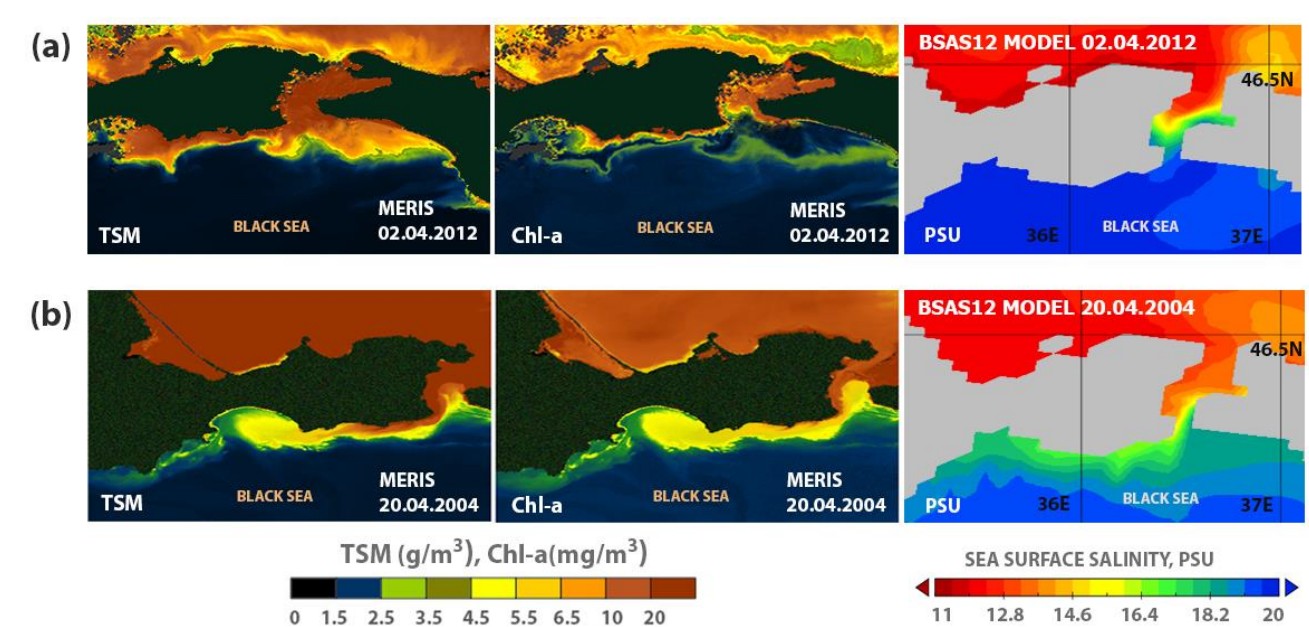

**Figure 2: Sea surface distributions of TSM (left) and Chl-a (middle) retrieved from MERIS satellite data and sea surface salinity distribution obtained from numerical modelling (right) indicating wind-induced resuspension of sea bottom sediments at the study area on 2 April 2012 (a) and spreading of AP on 20 April 2004 (b)**

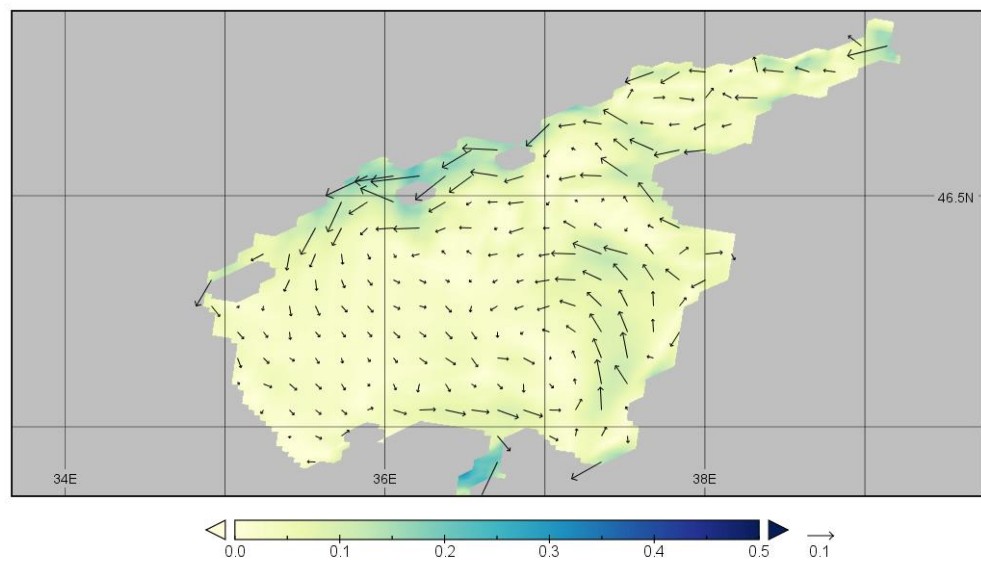

**Figure 3: Daily snapshot of surface velocity field on 15 January 1994. It illustrates the cyclonic Rim Current, the quasi-permanent anticyclonic currents - (Batumi, Sochi, Sebastopol), and the westward outflow from the Kerch Strait spreading along the Crimea peninsula.**

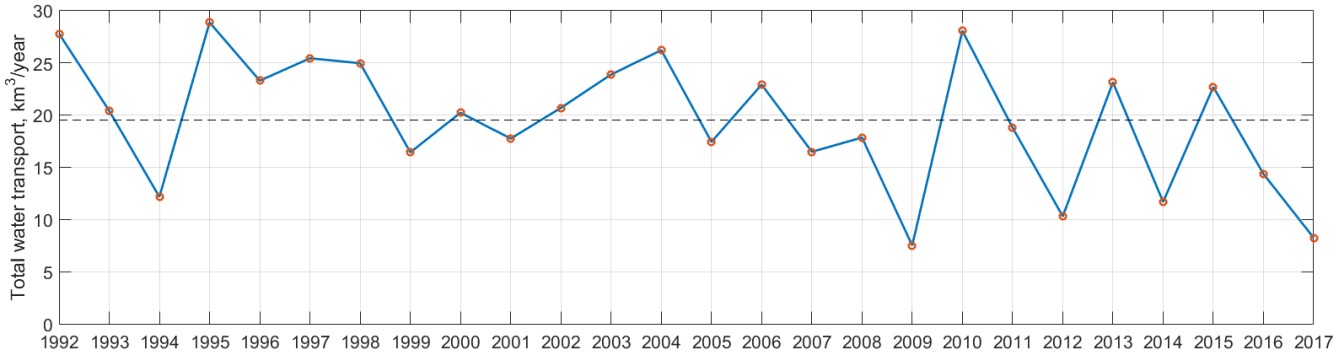

**Figure 4: Yearly mean values of total water transport through the Kerch Strait, km³. Dashed line - average yearly transport over the entire period of the experiment.**

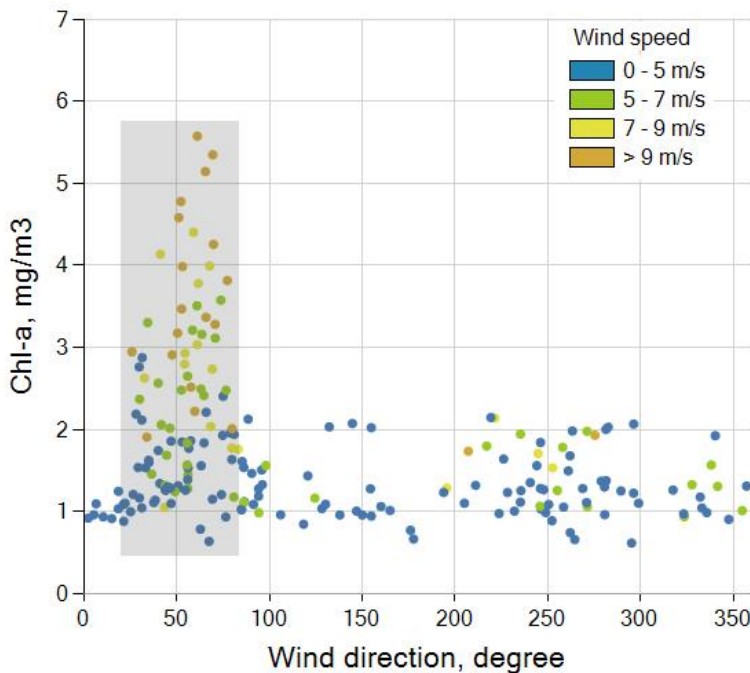

**Figure 5: Dependence between concentration of Chl-a in the Kerch Strait and the adjacent area of the Black Sea retrieved from optical satellite imagery, on the one hand, and direction of wind forcing averaged over 24 hours preceding satellite observations, on the other hand**

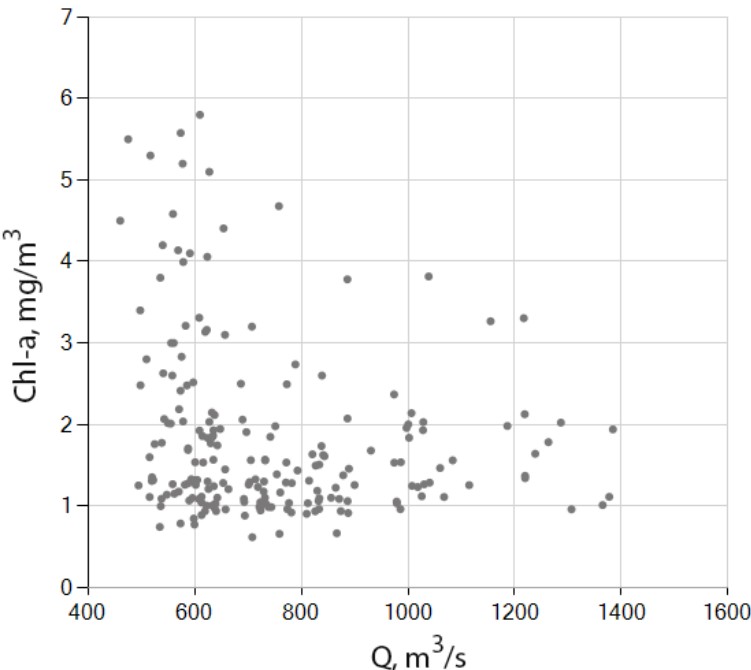

**Figure 6: Dependence between concentration of Chl-a in the Kerch Strait and the adjacent area of the Black Sea retrieved from optical satellite imagery, on the one hand, and total discharge of the Don and Kuban rivers, on the other hand.**

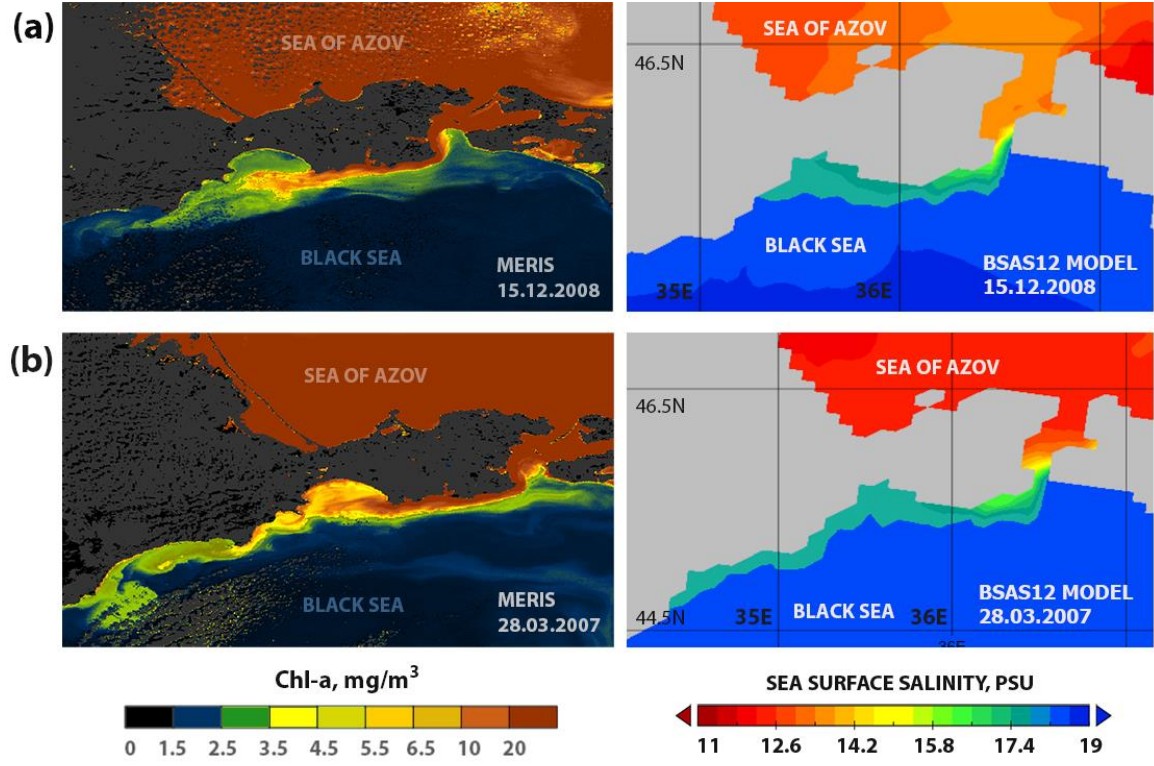

**Figure 7: Sea surface distribution of Chl-a retrieved from MERIS satellite data (left) and sea surface salinity distribution obtained from numerical modelling (right) on 12 December 2008 (a) and 28 March 2007 (b).**

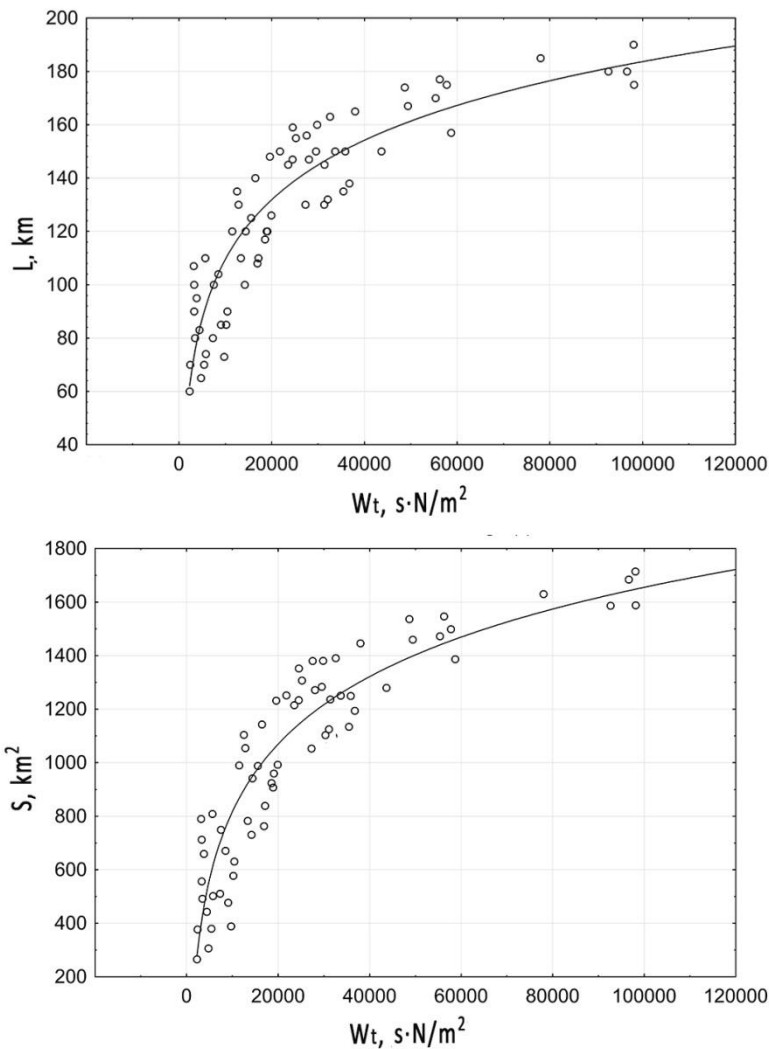

**Figure 8: Dependence between alongshore extent of AP (top) and area of AP (bottom) calculated from satellite data, on the one hand, and wind forcing index for the period of predominant northeasterly winds preceding time of satellite observations, on the other hand**

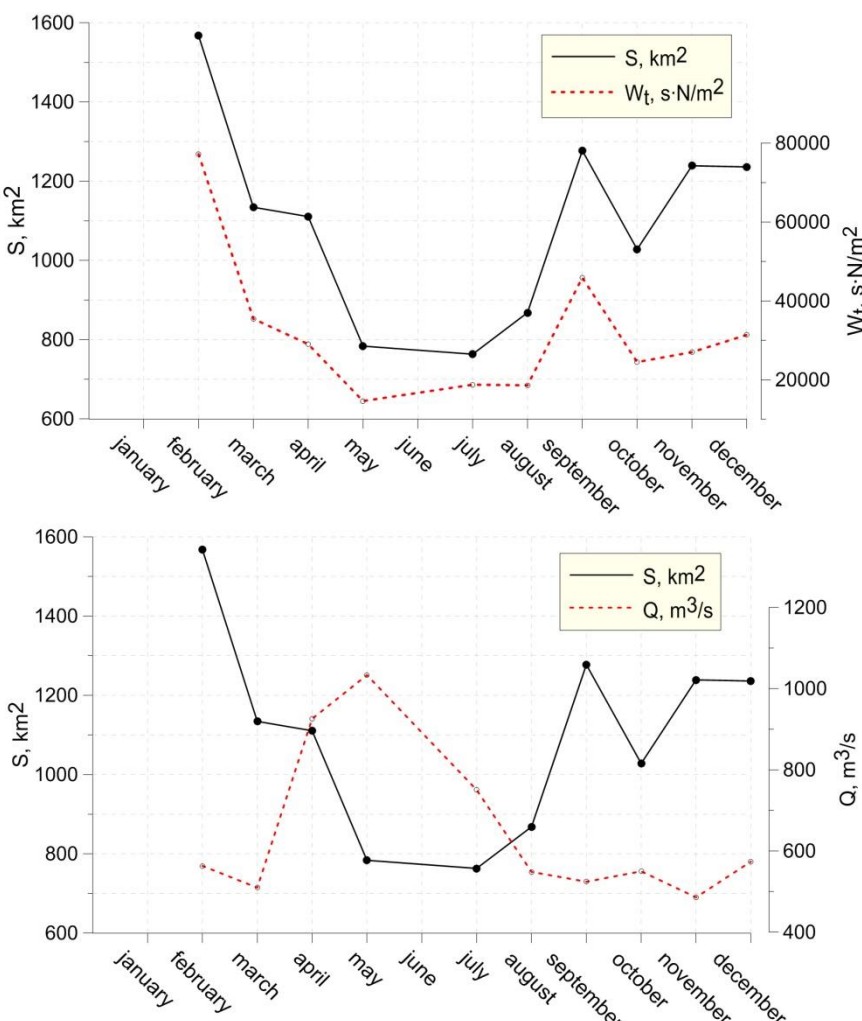

**Figure 9: Dependence between monthly averages of wind index (top) and total discharge rate of the Don and Kuban rivers (bottom), on the one hand, and monthly averages of area of AP detected at optical satellite imagery, on the other hand.**

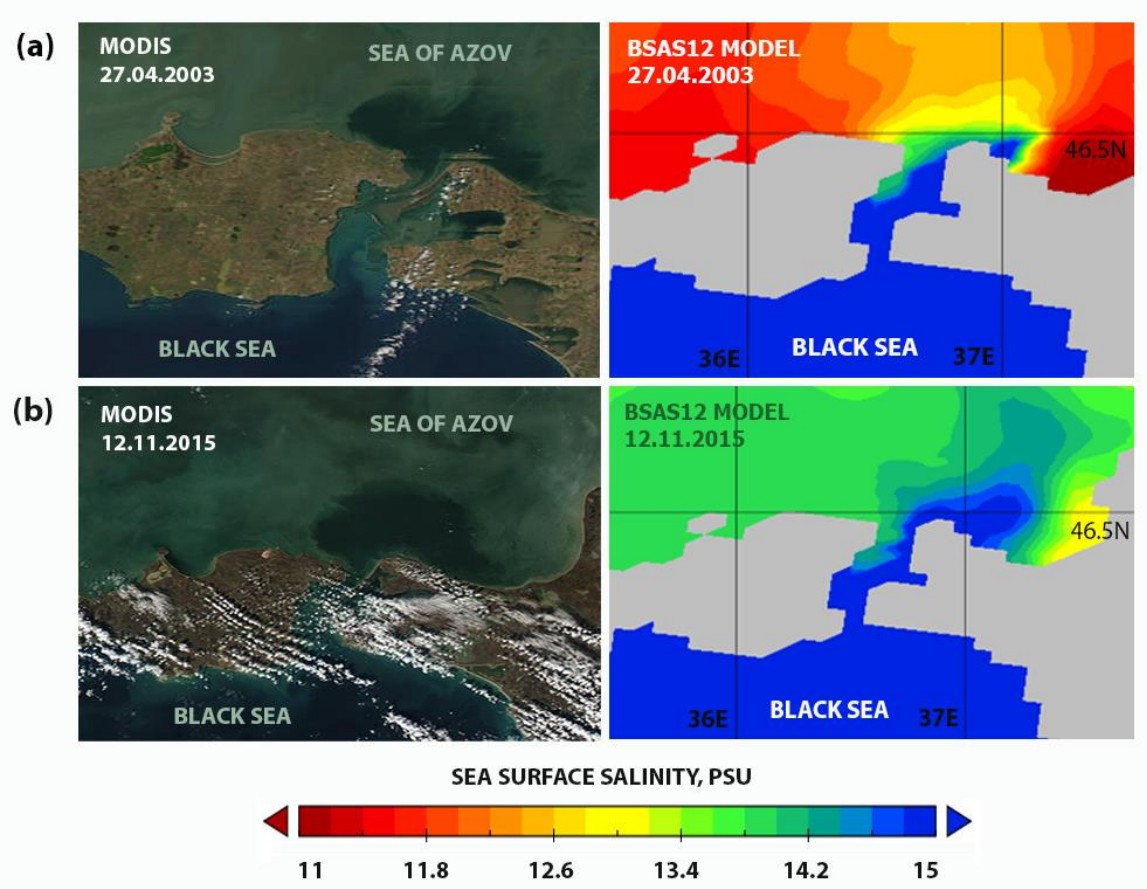

**Figure 10: Formation of the BP after the wind relaxation on 27 April 2003 (a) and 12 November 2015 (b), retrieved from MERIS satellite data and confirmed by the BSAS12 model simulation.**

**Table 1: The flow direction statistics in the Kerch Strait for the period from 1992 to 2010, obtained by the BSAS12 numerical model simulation.**

| Year | Only into Black sea, days | Only into Sea of Azov, days | In both directions, days |
|------|---------------------------|-----------------------------|--------------------------|
| 1992 | 178 | 145 | 43 |
| 1993 | 177 | 152 | 36 |
| 1994 | 169 | 150 | 46 |
| 1995 | 180 | 150 | 35 |
| 1996 | 171 | 153 | 42 |
| 1997 | 181 | 150 | 34 |
| 1998 | 173 | 146 | 46 |
| 1999 | 171 | 158 | 36 |
| 2000 | 168 | 156 | 42 |
| 2001 | 167 | 150 | 48 |
| 2002 | 170 | 150 | 45 |
| 2003 | 181 | 139 | 45 |
| 2004 | 174 | 152 | 40 |
| 2005 | 172 | 145 | 48 |
| 2006 | 167 | 146 | 52 |
| 2007 | 161 | 154 | 50 |
| 2008 | 176 | 149 | 41 |
| 2009 | 166 | 145 | 54 |
| 2010 | 163 | 157 | 45 |

