# Peer review of "Water exchange between the Sea of Azov and the Black Sea through the Kerch Strait"

_Ocean Science, 2019_

## Referee Comment (RC1) · Anonymous Referee #1 · 18 Mar 2019

The premise of this paper is a study the water exchange through the Kerch Strait. My assessment is that the authors rather studied sediment resuspension due to wind/wave events which is insufficient to infer the true water exchange that also exists without sediment addition.

Sediment resuspension is highly variable, but this should not be used to claim that the water exchange is too. More specific ocean data with mooring measurements and/or calibrated hydrodynamic modelling would be better suited to quantify this water exchange. Without such, most of the paper remains pure speculation and is of little scientific value.

[Figure]

A starting point should be to explore the seasonal cycle of density differences across the strait to identify any possible seasonality of the forcing. The existence of a density gradient (in conjunction with a barotropic pressure gradient) should induce a steady exchange circulation typical for positive estuaries. Indeed the water exchange through the mouth of a positive estuary is typically a two-way process with in- and outflows happening at the same time. Indeed, this exchange may be weaker or enhanced by winds as demonstrated in several studies before. To claim that winds are the sole initiator of episodic exchange flows is not aligned with previous studies on density-driven exchange flows in positive estuaries. The findings are heavily biased and misguided by the highly episodic nature of sediment resuspension events. To this end, I can only recommend this paper to be rejected for publication.

Other points:

Section 2: Give information on seasonal variations of density and differences across the Strait. Page 4, line 21: A persistent residual tidal flow of ∼5 cm/s could move substantial volumes of water through the strait as compared with rare episodes. Page 7: The authors claim that the area of an AP can substantially increase over time. In fact, it's the effect of mesoscale eddies (as obvious from the images) and the associated entrainment that causes this effect. However, this feature does NOT imply any increase in freshwater discharge rate.

---

## Referee Comment (RC2) · Anonymous Referee #2 · 29 Mar 2019

In this paper, the authors consider the Sea of Azov (SA) as an estuary and investigate a flow exchange between SA and the adjacent Black Sea (BS) by means of remote sensing. As a proxy for the buoyant outflow from SA into BS they use Cl-a, although on page 5 (lines 22-25) they state that SST, Cl-a and TSM "are prone to significant variability and/or do not act as passive tracers, which hinders their direct application for accurate identification" of buoyant outflow from SA. As a proxy for the BS water inflow into SA, the authors use TSM, which they have discarded as a reliable marker for buoyant water just one page earlier (i.e., page 7, lines 5-8). They have to rely on TSM signal now, because the BC inflow propagates near the bottom and does not produce immediate signature on the surface. The major conclusion of this study is

that the buoyant outflow from SA into BS occurs only under the external forcing by northeasterly (NE) winds, while inflow of heavy BS water into SA occurs during the relaxation of NE winds (e.g., page 10, line 24) and under any wind conditions. The authors also attempt to quantify "intensity" of the flow exchange through the Kerch Strait by determining the area of the SA plume in BS, and scaling this area against the integral of wind speed when the wind is favorable for the plume formation. In my opinion, there is a major confusion in this conclusion: do the authors imply that NE winds precondition the BS inflow into SA (that is, relaxation of NE winds triggers the BS inflow) or do they mean that the BS inflow occurs always as long as NE winds do not operate? The authors seem to interchange these two very different messages in different parts of their manuscript (e.g., compare lines 24-26 on page 10 and lines 23-26 on page 11). I feel strongly that the estuarine exchange flow should occur in both directions, but the authors seem to imply that under light or no wind forcing conditions, it is a one-way traffic: BS water flows in SA in a unidirectional manner. This is a bold and unsubstantiated claim. Frankly, I don't think that TSM signal can provide any reliable information about the presence of BS water at the bottom, it just tells us about the wind-induced resuspension of sediments. Apart from this major issue with paper's conclusions, I have some questions with the analysis. I am not sure why the authors use logarithms of properties compared in Figure 6 (and not the properties themselves). The bottom panel scatterplot does not resemble a linear function, so the linear regression is a poor choice here. This relationship is by no means linear: first, the wind driven transport is proportional to the wind stress (which is quadratic in wind speed). Second, the surface area is not directly proportional to the discharge Q through the Kerch Strait, but rather to Q/h, where h is the sickness of the plume, which is generally unknown but does depend on the wind stress. Regarding the upper panel of Figure 6, I think the alongshore extension of the plume under the downwelling wind forcing primarily depends on the ALONG-SHELF wind stress component (not the wind velocity magnitude). Throughout the paper, there is unfortunate confusion about the wind direction. In geosciences, wind direction indicates where the wind blows from and

it is measured from true north in clockwise direction. Wind direction 210 to 260 degrees (page 6, line 17) means that the wind blows from southwest, and it's not what the authors assume. Likewise, "northeastern" (wind) and "northeastward" have opposite meanings, northeastward denotes a flow from southwest towards northeast, but the authors freely interchange these two terms (e.g., lines 24 and 26 on page 10, and many other instances). On page 11 (2nd paragraph), the authors compare the SA volume and the annual volume of the freshwater discharge. This is a very strange proposition because a year is not a proper time scale for the exchange processes between AS and BC, according to the authors. In my opinion, the authors should rectify their arguments in this part of the discussion. Finally, the language should be checked throughout. For instance, it is a "boundary condition", not a "border condition". Word "significant" is badly overused. In scientific literature, this word tends to have a statistical context (that is, significant in quantitative, statistical sense). "Substantial" would probably sound better and less irritating for quantitatively minded readers. In conclusion, I recommend this manuscript to be returned to the authors for major revisions. I think the analysis of Cl-a can be salvaged, but it is unclear at this point if it will account for the full publishable unit.

---

## Referee Comment (RC3) · Anonymous Referee #2 · 29 Mar 2019

Of course it's "thickness" (of the plume), not "sickness". My apologies for this typo!

---

## Referee Comment (RC4) · Anonymous Referee #3 · 2 Apr 2019

The purpose of this paper was to analyze water exchange between Sea of Azov and Black sea. For this analyze author used satellite images and wind data. The author use TSM and CHL-a characteristics to trace spreading of AP and BP in the area. However on the page 5, author claims that: "above mentioned characteristics are prone to significant variability and not act as passive tracers". According to this paper AI events are induced by northeastern wind events, with speed >5 m/s, and during all other wind conditions the unidirectional flow towards the north occur. I would agree that during some exceptional events unidirectional flow can take place, but would suggest that water exchange in estuary should be two-way process, with simultaneous outflow and inflow events. As regards spreading of BP in the Sea of Azov using TSM satellite im-

ages, seems very unreliable information. As well as TSM was mentioned as not reliable tracer. Rather it can show wind-induced resuspension of sea-bottom sediments. In my opinion in present stage, presented paper does not give correct or reliable information regarding water exchange between the Sea of Azov and Black Sea. However, current simulations by operative model or some additional measurements could increase value to this paper. Additional questions to the author: 1. As it is mentioned in the paper, circulation in the surface layer in the northeastern part of the Black Sea is dominated by a westward current along the continental slope and an anticyclonic eddy, which is regularly formed between this current and the coast near the Kerch Strait. How big influence eddy have on spreading the freshwater from the Sea of Azov. 2. On the page 6, author mentions that they use Chl-a as a stable tracer for AP in the Black Sea. Does this mean, that satellite images of Chl-a can be used only during big concentrations of Chl-a, and for example in winter periods this method cannot be used?

---

## Author Comment (AC2) · 7 Jun 2019

We strongly appreciate the reviewers' suggestions and comments that served to improve the article. In response to them, first, we added results of numerical modelling that supported the results obtained from analysis of satellite imagery. Second, we clarified methods of detection of surface-advected outflows from the Sea of Azov to the Black Sea and bottom-advected inflows in the opposite direction based on agreement between Chl-a and TSM satellite imagery. Third, we modified the interpretation of influence of wing forcing on inflows from the Black Sea to the Sea of Azov and provided a more thorough physical background based on both satellite data and numerical modelling. Finally, we adopted many specific comments and recommendations from the reviewers, reworked the structure of the manuscript, and improved the quality of English text and figures. The detailed reply to reviewers comments and description of changes made in the manuscript are given below and in the attached file.

1. In this paper, the authors consider the Sea of Azov (SA) as an estuary and investigate a flow exchange between SA and the adjacent Black Sea (BS) by means of remote sensing. As a proxy for the buoyant outflow from SA into BS they use Cl-a, although on page 5 (lines 22-25) they state that SST, Cl-a and TSM "are prone to significant variability and/or do not act as passive tracers, which hinders their direct application for accurate identification" of buoyant outflow from SA.

Response:

Thank you for this valuable comment. SST, Chl-a, and TSM indeed are prone to significant variability and straightforward usage of these individual characteristics for detection of outflow from SA to BS can be misleading. Thus, in the revised version of the manuscript we used more comprehensive joint analysis of SST, Chl-a, and TSM. Inflow events from SA to BS were identified by elevated concentration of Chl-a in the Kerch Strait and the adjacent coastal area of the Black Sea, because Chl-a has the lowest short-term variability among the considered sea surface characteristics and are not affected by wind/wave-induced sediment resuspension. If an inflow event was detected, we analyzed areas of elevated TSM, Chl-a, and elevated (in summer) or reduced (in winter) SST associated with formation of the Azov plume (AP) in the northeastern part of the Black Sea. If general forms and spatial scales of these areas were similar, we defined borders of AP basing on gradient of TSM that is the most stable passive tracer of AP in absence of episodic wind-induced bottom resuspension events. If areas of TSM, Chl-a, and SST anomalies were not consistent with each other, we assumed that areas of elevated TSM and reduced SST (in winter) were modified by wind-induced resuspension and mixing of AP with subjacent sea. In this case we defined borders of AP basing on gradient of Chl-a that is the most stable tracer of AP during intense wind

forcing conditions. These issues were described in the text. Also we applied a calibrated hydrodynamic modelling using the BSAS12 model to prove the results obtained from the analysis of satellite imagery. Salinity distributions simulated by BSAS12 in the study region showed good agreement with detection of AP and BP at satellite imagery, that is explicitly described in the revised version of the manuscript and illustrated by Figures 2, 8, 11. Thus, the applied method of identification of AP and BP and the obtained results are supported by both satellite data and numerical modelling.

2. As a proxy for the BS water inflow into SA, the authors use TSM, which they have discarded as a reliable marker for buoyant water just one page earlier (i.e., page 7, lines 5-8). They have to rely on TSM signal now, because the BS inflow propagates near the bottom and does not produce immediate signature on the surface. Frankly, I don't think that TSM signal can provide any reliable information about the presence of BS water at the bottom, it just tells us about the wind-induced resuspension of sediments.

Response:

TSM is not a reliable marker of SA water in the Black Sea because TSM increases over shallow areas in case of resuspension of bottom sediments. Thus, areas of elevated TSM can be induced by both sediment resuspension and spreading of SA water in the Black Sea. However, spreading of BS water in the Sea of Azov is indicated by reduced values of TSM in the shallow area adjacent to the Kerch Strait. Thus, resuspension of bottom sediments which increases TSM at the study region can not be misleading for detection of SA water indicated by reduced TSM especially at shallow areas. Thus, TSM is not a reliable marker of SA water in the Black Sea, but is a reliable marker of BS water in the Sea of Azov. Moreover, in the revised version we analyzed both TSM and Chl-a, and identified SA water in the Black Sea as areas of reduced TSM and Chl-a. This procedure provides more confidence in detection of SA water, as compared to usage of only TSM. As a result, the scheme of identification of inflow events from the Black Sea to the Sea of Azov and detection of borders of the bottom-advected Black Sea plume (BP) at satellite imagery is relatively straightforward, as compared

to identification of inflow events from the Sea of Azov to the Black Sea. However, we assume that BP is not manifested by anomalies of TSM and Chl-a in the surface layer during low wind forcing conditions, while during strong wind forcing conditions resuspension of bottom sediments can induce elevated concentrations of TSM in the surface layer that also hinders identification of BP. As a result, many of BI events are not detected by optical satellite imagery.

3. The major conclusion of this study is that the buoyant outflow from SA into BS occurs only under the external forcing by northeasterly (NE) winds, while relaxation of NE winds occurs during the relaxation of NE winds (e.g., page 10, line 24) and under any wind conditions. The authors also attempt to quantify "intensity" of the flow exchange through the Kerch Strait by determining the area of the SA plume in BS, and scaling this area against the integral of wind speed when the wind is favorable for the plume formation. In my opinion, there is a major confusion in this conclusion: do the authors imply that NE winds precondition the BS inflow into SA (that is, relaxation of NE winds triggers the BS inflow) or do they mean that the BS inflow occurs always as long as NE winds do not operate? The authors seem to interchange these two very different messages in different parts of their manuscript (e.g., compare lines 24-26 on page 10 and lines 23-26 on page 11).

Response:

Many thanks for this important comment. In the revised version of the manuscript we clarified meaning of "relaxation of NE winds" related to inflow of BS waters into SA. We state that inflow of BS waters into SA occurs after reverse of a barotropic pressure gradient as a result of relaxation of NE winds, i.e. relaxation of NE winds triggers the BS inflow. This result was supported by numerical modelling.

4. I feel strongly that the estuarine exchange flow should occur in both directions, but the authors seem to imply that under light or no wind forcing conditions, it is a one-way traffic: BS water flows in SA in a unidirectional manner. This is a bold and

unsubstantiated claim.

Response:

Many thanks for this important comment. We totally agree that water exchange through the mouth of a positive estuary is typically a two-way process with in- and outflows happening at the same time. However, it is not the case of narrow and shallow central part of the Kerch Strait, which width and depth are equal to 3 km and 3-4 m. In absence of external forcing water exchange through the Kerch Strait indeed is a two-way process. However, due to the dominating role of a barotropic component in the total pressure gradient along the strait (Figure 5 in the revised version of the manuscript), even moderate wind forcing at the study region induces a one-way water exchange in the narrowest part of the strait which defines water exchange between the Azov and Black seas. Numerical simulations show that a two-way water exchange occurred in the Kerch Strait only during weak wind forcing conditions which total annual duration was 34 – 54 days in 1992 – 2010, which is only 9-15% of the whole year (Table 1). Thus, one-way water exchange was observed during the majority of the year. This result is also supported by previous in situ observations (Ivanov, 2011) and numerical modelling (Stanev et al., 2017) of water exchange in the Kerch Strait. The rest of the year a one-way water exchange was observed. This point was described in the text.

5. Apart from this major issue with paper's conclusions, I have some questions with the analysis. I am not sure why the authors use logarithms of properties compared in Figure 6 (and not the properties themselves). The bottom panel scatterplot does not resemble a linear function, so the linear regression is a poor choice here. This relationship is by no means linear: first, the wind driven transport is proportional to the wind stress (which is quadratic in wind speed). Second, the surface area is not directly proportional to the discharge Q through the Kerch Strait, but rather to Q/h, where h is the sickness of the plume, which is generally unknown but does depend on the wind stress.

Response:

We agree that scatterplots in this figure (Figure 9 in the revised version of the manuscript) do not resemble linear functions, so the linear regression is a poor choice. In the revised version of the manuscript we improved these dependences and their approximations. Alongshore extent and area of AP increase with increase of wind forcing index. However, this increase is not steady, its derivative decreases with increase of wind forcing index. In particular if wind forcing index exceeds 3500 km, alongshore extent and area of AP are almost stable. Thus, the observed forms of dependences between have good approximation by logarithmic functions. We are not aware that logarithmic approximation has any straightforward physical background (despite steady decrease of dependence derivative) due to complex and strongly non-linear dependences between wind speed, discharge through the Kerch Strait, and spatial characteristics of AP. However, the obtained relations are essential for numerical parameterizations of water exchange through the Kerch Strait based on wind data, which are addressed in this study.

6. Regarding the upper panel of Figure 6, I think the alongshore extension of the plume under the downwelling wind forcing primarily depends on the ALONG-SHELF wind stress component (not the wind velocity magnitude).

Response:

We agree with this point. Spatial scales of a buoyant plume in a non-tidal sea are defined mainly by two external forcing parameters, namely, river discharge rate and wind forcing. Thus, alongshore extent of AP depends on discharge rate of SA water through the Kerch Strait and local wind forcing. Inflow of water from SA to BS and formation of AP occurs only under north-easterly wind forcing. As a result, cross-shelf wind component determines discharge rate of SA water through the Kerch Strait and both wind components govern further alongshore extension of AP. In this study, we consider alongshore extension and area of AP only during periods of inflow from SA to

BS, i.e., we do not consider AP detached from its source in the Kerch Strait after reverse of flow direction in the strait and cessation of inflow from SA to BS. Thus, we calculate alongshore extension and area of AP only during moderate and strong north-easterly winds that induce inflow from SA to BS. During these periods, alongshore extension of AP indeed shows good relation with variability both cross- shelf and along-shelf wind component. However, the dependence of alongshore extension of AP on wind velocity magnitude is better due to small range of the considered wind directions. This issue was clarified in the text.

7. Throughout the paper, there is unfortunate confusion about the wind direction. In geosciences, wind direction indicates where the wind blows from and it is measured from true north in clockwise direction. Wind direction 210 to 260 degrees (page 6, line 17) means that the wind blows from southwest, and it's not what the authors assume. Likewise, "northeastern" (wind) and "northeastward" have opposite meanings, northeastward denotes a flow from southwest towards northeast, but the authors freely interchange these two terms (e.g., lines 24 and 26 on page 10, and many other instances). Response: Thank you for this point, we checked all parts of the manuscript where wind direction was mentioned and corrected the wrong usage of the related words.

8. On page 11 (2nd paragraph), the authors compare the SA volume and the annual volume of the freshwater discharge. This is a very strange proposition because a year is not a proper time scale for the exchange processes between AS and BS, according to the authors. In my opinion, the authors should rectify their arguments in this part of the discussion.

Response:

We agree with the reviewer that we should improve arguments in this part of the discussion. In this paragraph we discuss relation between freshwater discharge to river estuary and water exchange between the estuary and open sea, i.e. does signal of

river discharge dissipate in the estuary or it influences freshwater outflow from estuary to open sea. This process depends on two main factors, namely, volume of inflowing river discharge and spatial scales of an estuary. If large river inflows to small estuary (e.g., the Amur River and the Amur Liman), variability of river discharge rate strongly affects water outflow from estuary to open sea. On the other hand, discharge of a small river inflowing to a large estuary does not influence and water exchange between the estuary and open sea. The area and volume of the Sea of Azov are large enough; therefore, signals of synoptic and even significant seasonal variability of discharge of the DON and Kuban rivers dissipate in the Sea of Azov and do not influence formation and/or intensity of AI events. It was clarified in the manuscript

9. Finally, the language should be checked throughout. For instance, it is a "boundary condition", not a "border condition". Word "significant" is badly overused. In scientific literature, this word tends to have a statistical context (that is, significant in quantitative, statistical sense). "Substantial" would probably sound better and less irritating for quantitatively minded readers.

Response:

Thank you for these points, we adopted them.

Please also note the supplement to this comment:
https://www.ocean-sci-discuss.net/os-2019-2/os-2019-2-AC2-supplement.pdf

**Supplement:**

**Reply to comments from the reviewers and summary of the major changes made in the revised manuscript OS-2018-2**

We strongly appreciate the reviewers' suggestions and comments that served to improve the article. In response to them,
5   first, we added results of numerical modelling that supported the results obtained from analysis of satellite imagery. Second, we clarified methods of detection of surface-advected outflows from the Sea of Azov to the Black Sea and bottom-advected inflows in the opposite direction based on agreement between Chl-a and TSM satellite imagery. Third, we modified the interpretation of influence of wing forcing on inflows from the Black Sea to the Sea of Azov and provided a more thorough physical background based on both satellite data and numerical modelling. Finally, we adopted many specific comments and
10   recommendations from the reviewers, reworked the structure of the manuscript, and improved the quality of English text and figures. The detailed reply to reviewers comments and description of changes made in the manuscript are given below.

**1. In this paper, the authors consider the Sea of Azov (SA) as an estuary and investigate a flow exchange between SA and the adjacent Black Sea (BS) by means of remote sensing. As a proxy for the buoyant outflow from SA into BS**
15   **they use Cl-a, although on page 5 (lines 22-25) they state that SST, Cl-a and TSM "are prone to significant variability and/or do not act as passive tracers, which hinders their direct application for accurate identification" of buoyant outflow from SA.**

Response:
20   Thank you for this valuable comment. SST, Chl-a, and TSM indeed are prone to significant variability and straightforward usage of these individual characteristics for detection of outflow from SA to BS can be misleading. Thus, in the revised version of the manuscript we used more comprehensive joint analysis of SST, Chl-a, and TSM. Inflow events from SA to BS were identified by elevated concentration of Chl-a in the Kerch Strait and the adjacent coastal area of the Black Sea, because Chl-a has the lowest short-term variability among the considered sea surface characteristics and are not affected by
25   wind/wave-induced sediment resuspension. If an inflow event was detected, we analyzed areas of elevated TSM, Chl-a, and elevated (in summer) or reduced (in winter) SST associated with formation of the Azov plume (AP) in the northeastern part of the Black Sea. If general forms and spatial scales of these areas were similar, we defined borders of AP basing on gradient of TSM that is the most stable passive tracer of AP in absence of episodic wind-induced bottom resuspension events. If areas of TSM, Chl-a, and SST anomalies were not consistent with each other, we assumed that areas of elevated TSM and reduced
30   SST (in winter) were modified by wind-induced resuspension and mixing of AP with subjacent sea. In this case we defined borders of AP basing on gradient of Chl-a that is the most stable tracer of AP during intense wind forcing conditions. These issues were described in the text. Also we applied a calibrated hydrodynamic modelling using the BSAS12 model to prove the results obtained from the analysis of satellite imagery. Salinity distributions simulated by BSAS12 in the study region showed good agreement with detection of AP and BP at satellite imagery, that is explicitly described in the revised version

of the manuscript and illustrated by Figures 2, 8, 11. Thus, the applied method of identification of AP and BP and the obtained results are supported by both satellite data and numerical modelling.

**2. As a proxy for the BS water inflow into SA, the authors use TSM, which they have discarded as a reliable marker for buoyant water just one page earlier (i.e., page 7, lines 5-8). They have to rely on TSM signal now, because the BS inflow propagates near the bottom and does not produce immediate signature on the surface.**
**Frankly, I don't think that TSM signal can provide any reliable information about the presence of BS water at the bottom, it just tells us about the wind-induced resuspension of sediments.**

Response:

TSM is not a reliable marker of SA water in the Black Sea because TSM increases over shallow areas in case of resuspension of bottom sediments. Thus, areas of elevated TSM can be induced by both sediment resuspension and spreading of SA water in the Black Sea. However, spreading of BS water in the Sea of Azov is indicated by reduced values of TSM in the shallow area adjacent to the Kerch Strait. Thus, resuspension of bottom sediments which increases TSM at the study region can not be misleading for detection of SA water indicated by reduced TSM especially at shallow areas. Thus, TSM is not a reliable marker of SA water in the Black Sea, but is a reliable marker of BS water in the Sea of Azov. Moreover, in the revised version we analyzed both TSM and Chl-a, and identified SA water in the Black Sea as areas of reduced TSM and Chl-a. This procedure provides more confidence in detection of SA water, as compared to usage of only TSM. As a result, the scheme of identification of inflow events from the Black Sea to the Sea of Azov and detection of borders of the bottom-advected Black Sea plume (BP) at satellite imagery is relatively straightforward, as compared to identification of inflow events from the Sea of Azov to the Black Sea. However, we assume that BP is not manifested by anomalies of TSM and Chl-a in the surface layer during low wind forcing conditions, while during strong wind forcing conditions resuspension of bottom sediments can induce elevated concentrations of TSM in the surface layer that also hinders identification of BP. As a result, many of BI events are not detected by optical satellite imagery.

**3. The major conclusion of this study is that the buoyant outflow from SA into BS occurs only under the external forcing by northeasterly (NE) winds, while relaxation of NE winds occurs during the relaxation of NE winds (e.g., page 10, line 24) and under any wind conditions. The authors also attempt to quantify "intensity" of the flow exchange through the Kerch Strait by determining the area of the SA plume in BS, and scaling this area against the integral of wind speed when the wind is favorable for the plume formation. In my opinion, there is a major confusion in this conclusion: do the authors imply that NE winds precondition the BS inflow into SA (that is, relaxation of NE winds triggers the BS inflow) or do they mean that the BS inflow occurs always as long as NE winds do not operate? The authors seem to interchange these two very different messages in different parts of their manuscript (e.g., compare lines 24-26 on page 10 and lines 23-26 on page 11).**

Response:

Many thanks for this important comment. In the revised version of the manuscript we clarified meaning of "relaxation of NE winds" related to inflow of BS waters into SA. We state that inflow of BS waters into SA occurs after reverse of a barotropic pressure gradient as a result of relaxation of NE winds, i.e. relaxation of NE winds triggers the BS inflow. This result was supported by numerical modelling.

**4. I feel strongly that the estuarine exchange flow should occur in both directions, but the authors seem to imply that under light or no wind forcing conditions, it is a one-way traffic: BS water flows in SA in a unidirectional manner. This is a bold and unsubstantiated claim.**

Response:

Many thanks for this important comment. We totally agree that water exchange through the mouth of a positive estuary is typically a two-way process with in- and outflows happening at the same time. However, it is not the case of narrow and shallow central part of the Kerch Strait, which width and depth are equal to 3 km and 3-4 m. In absence of external forcing water exchange through the Kerch Strait indeed is a two-way process. However, due to the dominating role of a barotropic component in the total pressure gradient along the strait (Figure 5 in the revised version of the manuscript), even moderate wind forcing at the study region induces a one-way water exchange in the narrowest part of the strait which defines water exchange between the Azov and Black seas. Numerical simulations show that a two-way water exchange occurred in the Kerch Strait only during weak wind forcing conditions which total annual duration was 34 – 54 days in 1992 – 2010, which is only 9-15% of the whole year (Table 1). Thus, one-way water exchange was observed during the majority of the year. This result is also supported by previous in situ observations (Ivanov, 2011) and numerical modelling (Stanev et al., 2017) of water exchange in the Kerch Strait. The rest of the year a one-way water exchange was observed. This point was described in the text.

**5. Apart from this major issue with paper's conclusions, I have some questions with the analysis. I am not sure why the authors use logarithms of properties compared in Figure 6 (and not the properties themselves). The bottom panel scatterplot does not resemble a linear function, so the linear regression is a poor choice here. This relationship is by no means linear: first, the wind driven transport is proportional to the wind stress (which is quadratic in wind speed). Second, the surface area is not directly proportional to the discharge Q through the Kerch Strait, but rather to Q/h, where h is the sickness of the plume, which is generally unknown but does depend on the wind stress.**

Response:

We agree that scatterplots in this figure (Figure 9 in the revised version of the manuscript) do not resemble linear functions, so the linear regression is a poor choice. In the revised version of the manuscript we improved these dependences and their approximations. Alongshore extent and area of AP increase with increase of wind forcing index. However, this increase is not steady, its derivative decreases with increase of wind forcing index. In particular if wind forcing index exceeds 3500 km, alongshore extent and area of AP are almost stable. Thus, the observed forms of dependences between have good approximation by logarithmic functions. We are not aware that logarithmic approximation has any straightforward physical background (despite steady decrease of dependence derivative) due to complex and strongly non-linear dependences between wind speed, discharge through the Kerch Strait, and spatial characteristics of AP. However, the obtained relations are essential for numerical parameterizations of water exchange through the Kerch Strait based on wind data, which are addressed in this study.

**6. Regarding the upper panel of Figure 6, I think the alongshore extension of the plume under the downwelling wind forcing primarily depends on the ALONG-SHELF wind stress component (not the wind velocity magnitude).**

Response:

We agree with this point. Spatial scales of a buoyant plume in a non-tidal sea are defined mainly by two external forcing parameters, namely, river discharge rate and wind forcing. Thus, alongshore extent of AP depends on discharge rate of SA water through the Kerch Strait and local wind forcing. Inflow of water from SA to BS and formation of AP occurs only under north-easterly wind forcing. As a result, cross-shelf wind component determines discharge rate of SA water through the Kerch Strait and both wind components govern further alongshore extension of AP. In this study, we consider alongshore extension and area of AP only during periods of inflow from SA to BS, i.e., we do not consider AP detached from its source in the Kerch Strait after reverse of flow direction in the strait and cessation of inflow from SA to BS. Thus, we calculate alongshore extension and area of AP only during moderate and strong north-easterly winds that induce inflow from SA to BS. During these periods, alongshore extension of AP indeed shows good relation with variability both cross- shelf and along-shelf wind component. However, the dependence of alongshore extension of AP on wind velocity magnitude is better due to small range of the considered wind directions. This issue was clarified in the text.

**7. Throughout the paper, there is unfortunate confusion about the wind direction. In geosciences, wind direction indicates where the wind blows from and it is measured from true north in clockwise direction. Wind direction 210 to 260 degrees (page 6, line 17) means that the wind blows from southwest, and it's not what the authors assume. Likewise, "northeastern" (wind) and "northeastward" have opposite meanings, northeastward denotes a flow from southwest towards northeast, but the authors freely interchange these two terms (e.g., lines 24 and 26 on page 10, and many other instances).**

Response:

Thank you for this point, we checked all parts of the manuscript where wind direction was mentioned and corrected the wrong usage of the related words.

5 **8. On page 11 (2nd paragraph), the authors compare the SA volume and the annual volume of the freshwater discharge. This is a very strange proposition because a year is not a proper time scale for the exchange processes between AS and BS, according to the authors. In my opinion, the authors should rectify their arguments in this part of the discussion.**

10 Response:

We agree with the reviewer that we should improve arguments in this part of the discussion. In this paragraph we discuss relation between freshwater discharge to river estuary and water exchange between the estuary and open sea, i.e. does signal of river discharge dissipate in the estuary or it influences freshwater outflow from estuary to open sea. This process depends on two main factors, namely, volume of inflowing river discharge and spatial scales of an estuary. If large river inflows to

15 small estuary (e.g., the Amur River and the Amur Liman), variability of river discharge rate strongly affects water outflow from estuary to open sea. On the other hand, discharge of a small river inflowing to a large estuary does not influence and water exchange between the estuary and open sea. The area and volume of the Sea of Azov are large enough; therefore, signals of synoptic and even significant seasonal variability of discharge of the DON and Kuban rivers dissipate in the Sea of Azov and do not influence formation and/or intensity of AI events. It was clarified in the manuscript

**9. Finally, the language should be checked throughout. For instance, it is a "boundary condition", not a "border condition". Word "significant" is badly overused. In scientific literature, this word tends to have a statistical context (that is, significant in quantitative, statistical sense). "Substantial" would probably sound better and less irritating for quantitatively minded readers.**

Response:

Thank you for these points, we adopted them.

**References**

[revised manuscript text omitted]
 ~~wind data described in Section 4.1 showed that northeastward wind forcing governs formation of AI events and spreading of AP on synoptic and seasonal time scales. Thus, we can reconstruct dependence of intensity of AI, i.e., discharge rate from the Sea of Azov to the Black Sea through the Kerch Strait, on speed and duration of local wind forcing. First, we define "AI favorable" wind conditions as a period of predominant northeastern wind forcing, which wind forcing index $T_s$ exceeds 500 km. All other periods are prescribed as "BI-favorable" wind conditions. Second, wind reanalysis for the study region, we identify periods of AI favorable wind events. Finally, we calculate areas of AP formed during these events using the obtained relation between wind forcing index and area of AP. Sum these areas for all AI events occurred during a year ($S_{AP}$) is indicative of total annual volume of water inflow from the Sea of Azov to the Black Sea through the Kerch Strait ($V_{AI}$). Simonov & Altman (1991) estimate mean value of $V_{AI}$ during 1963-1972 as 64.3 km³, while we calculated that mean value of $S_{AP}$ during 1963-1972 is equal to 12747 km². We assume that mean depth of AP ($H_{AP}$) does not significantly depend on its spatial scale and is mainly defined by bathymetry of the Kerch Strait. Thus, we obtain that $V_{AI} = H_{AP} \cdot S_{AP}$ and mean annual depth of AP is $H_{AP} = V_{AI} / S_{AP} = 5$ m, which is consistent with depth of the Kerch Strait (Figure 1). Finally, using the dependence of plume area on wind forcing index and estimation of mean depth of AP we obtain the following equation for water inflow volume during an AI event ($V$): $V = S \cdot H_{AP} = 0.017 \cdot W_t^{-0.77}$.
Basing on the equations described above, we estimated mean monthly and annual frequency, duration, and share of AI favorable and BI favorable wind conditions during the period of 2000-2017 and calculated volume of discharge from the Sea of Azov to the Black Sea (table 1). The most frequent and long AI favorable wind conditions are registered in autumn and winter. AI events occur during 36-39% of days in these months except a distinct peak in October, when they account for 52% of days. Then their frequency decreases from 2.2-3.1 times a month in September – February to 0.9-1.2 times a month in May – July that results in decrease of share of AI events to 11-15% of days during the latter period. On the other hand, mean monthly duration of AI favorable wind conditions is stable during the year and varies from 91.4 hour in June to 119.1 in October. BI favorable wind conditions, on the opposite, are dominating in May – July (85-89% of days) and then their share steadily decreases till October (48% of days). Frequency of BI events is the same as frequency of AI events, while mean monthly duration of BI events shows significantly greater seasonal variability from 135 hours in October to 718.5 in~~

**Conclusions**

[revised manuscript text omitted]

~~In this study we revealed wind forcing conditions that cause formation of AI and BI events and analyzed intra annual variability of their monthly and annual frequency and duration for the period of 2000-2017. BI favorable wind conditions are significantly more frequent than AI favorable conditions and occur during 71% of days in a year with a distinct peak in May – July (85-89% of days). AI events slightly dominate over BI events only in October (58% of days), while during the rest of the year their share is 11-39%. Shifts between AI favorable and BI favorable wind conditions occur relatively rarely, 1-3 times in a month and only 46 times in a year. Thus, AI and BI events generally last for relatively long time periods; their mean annual durations are approximately 4.3 and 13.7 days, respectively. Duration of AI events does not significantly change during a year, while mean monthly duration of BI event varies from 5 to 30 days.~~

~~Also we obtained relations between spatial characteristics of AP, namely, alongshore scale and area, on wind forcing conditions during AI event. On the other hand, we did not determine relevant equations for water transport from the Black Sea to the Sea of Azov. It is caused by the fact that spatial characteristics and temporal variability of spreading of BP in the bottom layer of the Sea of Azov is much worse detected at satellite imagery, as compared to surface-advected AP. As a result, definition of dependencies for both discharge rate from the Black Sea to the Sea of Azov and spatial characteristics of BP on wind forcing requires specific in situ measurements which are beyond the current study.~~

[revised manuscript text omitted]

---

## Author Comment (AC3) · 7 Jun 2019

We strongly appreciate the reviewers' suggestions and comments that served to improve the article. In response to them, first, we added results of numerical modelling that supported the results obtained from analysis of satellite imagery. Second, we clarified methods of detection of surface-advected outflows from the Sea of Azov to the Black Sea and bottom-advected inflows in the opposite direction based on agreement between Chl-a and TSM satellite imagery. Third, we modified the interpretation of influence of wing forcing on inflows from the Black Sea to the Sea of Azov and provided a more thorough physical background based on both satellite data and numerical mod-

elling. Finally, we adopted many specific comments and recommendations from the reviewers, reworked the structure of the manuscript, and improved the quality of English text and figures. The detailed reply to reviewers' comments and description of changes made in the manuscript are given below and in the attached file.

1. The purpose of this paper was to analyze water exchange between Sea of Azov and Black sea. For this analyze author used satellite images and wind data. The author use TSM and CHL-a characteristics to trace spreading of AP and BP in the area. However on the page 5, author claims that: "above mentioned characteristics are prone to significant variability and not act as passive tracers".

Response:

Thank you for this valuable comment. In the revised version of the manuscript we used joint analysis of sea surface temperature (SST), concentrations of chlorophyll a (Chl-a) and total suspended matter (TSM) retrieved from optical satellite data. These characteristics are prone to significant variability and not act as passive tracers. Thus these characteristics cannot be used for straightforward identification of inflow of freshened waters from the Sea of Azov to the Black Sea. However, these processes and their temporal scales are different for SST (diurnal cycle of solar radiation), TSM (episodic wind-induced bottom resuspension events), and Chl-a (synoptic and seasonal biological cycles). Thus, joint analysis of SST, TSM, and Chl-a distributions can be used for accurate detection of spreading of AP in the Black Sea. We applied the following scheme of identification of AI events and detection of borders of AP basing on satellite data. Inflow events were identified by elevated concentration of Chl-a in the Kerch Strait and the adjacent coastal area of the Black Sea, because Chl-a has the lowest short-term variability among the considered sea surface characteristics and are not affected by wind/wave-induced sediment resuspension. If an inflow event was detected, we analyzed areas of elevated TSM, Chl-a, and elevated (in summer) or reduced (in winter) SST associated with formation of AP in the northeastern part of the Black Sea. If general forms and spatial scales of these areas were similar, we defined borders of

AP basing on gradient of TSM that is the most stable passive tracer of AP in absence of episodic wind-induced bottom resuspension events. If areas of TSM, Chl-a, and SST anomalies were not consistent with each other, we assumed that areas of elevated TSM and reduced SST (in winter) were modified by wind-induced resuspension and mixing of AP with subjacent sea. In this case we defined borders of AP basing on gradient of Chl-a that is the most stable tracer of AP during intense wind forcing conditions. These issues were described in the text. Also we applied calibrated hydrodynamic modelling using the BSAS12 model to prove the results obtained from the analysis of satellite imagery. Salinity distributions simulated by BSAS12 in the study region showed good agreement with detection of AP and BP at satellite imagery, that is explicitly described in the revised version of the manuscript and illustrated by Figures 2, 8, 11. Thus, the applied method of identification of AP and BP and the obtained results are supported by both satellite data and numerical modelling.

2. According to this paper AI events are induced by northeastern wind events, with speed >5 m/s, and during all other wind conditions the unidirectional flow towards the north occur. I would agree that during some exceptional events unidirectional flow can take place, but would suggest that water exchange in estuary should be two-way process, with simultaneous outflow and inflow events.

Response:

Many thanks for this important comment. We totally agree that water exchange through the mouth of a positive estuary is typically a two-way process with in- and outflows happening at the same time. However, it is not the case of narrow and shallow central part of the Kerch Strait, which width and depth are equal to 3 km and 3-4 m. In absence of external forcing water exchange through the Kerch Strait indeed is a two-way process. However, due to the dominating role of a barotropic component in the total pressure gradient along the strait (Figure 5 in the revised version of the manuscript), even moderate wind forcing at the study region induces a one-way water exchange in the narrowest part of the strait which defines water exchange between the Azov and

Black seas. Numerical simulations show that a two-way water exchange occurred in the Kerch Strait only during weak wind forcing conditions which total annual duration was 34 – 54 days in 1992 – 2010, which is only 9-15% of the whole year (Table 1). This result is also supported by previous in situ observations (Ivanov, 2011) and numerical modelling (Stanev et al., 2017) of water exchange in the Kerch Strait. Thus, one-way water exchange was observed during the majority of the year. This point was described in the text.

3. As regards spreading of BP in the Sea of Azov using TSM satellite images, seems very unreliable information. As well as TSM was mentioned as not reliable tracer. Rather it can show wind-induced resuspension of sea-bottom sediments.

Response:

TSM is not a reliable marker of SA water in the Black Sea because TSM increases over shallow areas in case of resuspension of bottom sediments. Thus, areas of elevated TSM can be induced by both sediment resuspension and spreading of SA water in the Black Sea. However, spreading of BS water in the Sea of Azov is indicated by reduced values of TSM in the shallow area adjacent to the Kerch Strait. Thus, resuspension of bottom sediments which increases TSM at the study region can not be misleading for detection of SA water indicated by reduced TSM especially at shallow areas. Thus, TSM is not a reliable marker of SA water in the Black Sea, but is a reliable marker of BS water in the Sea of Azov. Moreover, in the revised version we analyzed both TSM and Chl-a and identified SA water in the Black Sea as areas of reduced TSM and Chl-a. This procedure provides more confidence in detection of SA water, as compared to usage of only TSM. Thus, the scheme of identification of BI events and detection of borders of BP at satellite imagery is relatively straightforward, as compared to identification of AI events. However, we assume that BP is not manifested by anomalies of TSM and Chl-a in the surface layer during low wind forcing conditions, while during strong wind forcing conditions resuspension of bottom sediments can induce elevated concentrations of TSM in the surface layer that also hinders identification of BP. As a

result, many of BI events are not detected by optical satellite imagery. Also salinity distributions simulated by BSAS12 in the study region showed good agreement with detection BP in the Sea of Azov at satellite imagery, that is explicitly described in the revised version of the manuscript and illustrated by Figure 11.

4. In my opinion in present stage, presented paper does not give correct or reliable information regarding water exchange between the Sea of Azov and Black Sea. However, current simulations by operative model or some additional measurements could increase value to this paper.

Response:

According to your recommendation we applied calibrated hydrodynamic modelling using the BSAS12 model to prove the results obtained from the analysis of satellite imagery. Salinity distributions simulated by BSAS12 in the study region showed good agreement with detection of AP and BP at satellite imagery, that is explicitly described in the revised version of the manuscript and illustrated by Figures 2, 8, 11. Thus, the applied method of identification of AP and BP and the obtained results are supported by both satellite data and numerical modelling.

5. As it is mentioned in the paper, circulation in the surface layer in the northeastern part of the Black Sea is dominated by a westward current along the continental slope and an anticyclonic eddy, which is regularly formed between this current and the coast near the Kerch Strait. How big influence eddy have on spreading the freshwater from the Sea of Azov.

Response:

Thank you for this point. Mesoscale eddies formed in the Black Sea near the Kerch Strait can largely influence spreading and mixing pattern of SA waters that inflow to the Black Sea and propagate westward along the southeastern shore of the Crimean Peninsula. Satellite data and numerical modelling reveal that AP are regularly entrained by mesoscale eddies that can significantly intensify cross-shelf transport of freshened water inflowing from the Sea of Azov. The resulting transport of freshened water to the deep part of the Black Sea can largely affect local biological and geochemical processes. This issue was discussed in the text, however, thorough study of this feature is beyond the scope of our study.

6. On the page 6, author mentions that they use Chl-a as a stable tracer for AP in the Black Sea. Does this mean, that satellite images of Chl-a can be used only during big concentrations of Chl-a, and for example in winter periods this method cannot be used?

Response:

Concentration of Chl-a in the Sea of Azov is larger than in the Black Sea during the whole year, including winter period. Thus, the applied method of detection of AP can be used for any periods of the year.

Please also note the supplement to this comment:
https://www.ocean-sci-discuss.net/os-2019-2/os-2019-2-AC3-supplement.pdf

**Supplement:**

**Reply to comments from the reviewers and summary of the major changes made in the revised manuscript OS-2018-2**

We strongly appreciate the reviewers' suggestions and comments that served to improve the article. In response to them,
5    first, we added results of numerical modelling that supported the results obtained from analysis of satellite imagery. Second, we clarified methods of detection of surface-advected outflows from the Sea of Azov to the Black Sea and bottom-advected inflows in the opposite direction based on agreement between Chl-a and TSM satellite imagery. Third, we modified the interpretation of influence of wing forcing on inflows from the Black Sea to the Sea of Azov and provided a more thorough physical background based on both satellite data and numerical modelling. Finally, we adopted many specific comments and
10    recommendations from the reviewers, reworked the structure of the manuscript, and improved the quality of English text and figures. The detailed reply to reviewers' comments and description of changes made in the manuscript are given below.

**1. The purpose of this paper was to analyze water exchange between Sea of Azov and Black sea. For this analyze author used satellite images and wind data. The author use TSM and CHL-a characteristics to trace spreading of AP**
15    **and BP in the area. However on the page 5, author claims that: "above mentioned characteristics are prone to significant variability and not act as passive tracers".**

Response:
Thank you for this valuable comment. In the revised version of the manuscript we used joint analysis of sea surface
20    temperature (SST), concentrations of chlorophyll a (Chl-a) and total suspended matter (TSM) retrieved from optical satellite data. These characteristics are prone to significant variability and not act as passive tracers. Thus these characteristics cannot be used for straightforward identification of inflow of freshened waters from the Sea of Azov to the Black Sea. However, these processes and their temporal scales are different for SST (diurnal cycle of solar radiation), TSM (episodic wind-induced bottom resuspension events), and Chl-a (synoptic and seasonal biological cycles). Thus, joint analysis of SST, TSM,
25    and Chl-a distributions can be used for accurate detection of spreading of AP in the Black Sea. We applied the following scheme of identification of AI events and detection of borders of AP basing on satellite data. Inflow events were identified by elevated concentration of Chl-a in the Kerch Strait and the adjacent coastal area of the Black Sea, because Chl-a has the lowest short-term variability among the considered sea surface characteristics and are not affected by wind/wave-induced sediment resuspension. If an inflow event was detected, we analyzed areas of elevated TSM, Chl-a, and elevated (in
30    summer) or reduced (in winter) SST associated with formation of AP in the northeastern part of the Black Sea. If general forms and spatial scales of these areas were similar, we defined borders of AP basing on gradient of TSM that is the most stable passive tracer of AP in absence of episodic wind-induced bottom resuspension events. If areas of TSM, Chl-a, and SST anomalies were not consistent with each other, we assumed that areas of elevated TSM and reduced SST (in winter) were modified by wind-induced resuspension and mixing of AP with subjacent sea. In this case we defined borders of AP

basing on gradient of Chl-a that is the most stable tracer of AP during intense wind forcing conditions. These issues were described in the text. Also we applied calibrated hydrodynamic modelling using the BSAS12 model to prove the results obtained from the analysis of satellite imagery. Salinity distributions simulated by BSAS12 in the study region showed good agreement with detection of AP and BP at satellite imagery, that is explicitly described in the revised version of the manuscript and illustrated by Figures 2, 8, 11. Thus, the applied method of identification of AP and BP and the obtained results are supported by both satellite data and numerical modelling.

**2. According to this paper AI events are induced by northeastern wind events, with speed >5 m/s, and during all other wind conditions the unidirectional flow towards the north occur. I would agree that during some exceptional events unidirectional flow can take place, but would suggest that water exchange in estuary should be two-way process, with simultaneous outflow and inflow events.**

Response:

Many thanks for this important comment. We totally agree that water exchange through the mouth of a positive estuary is typically a two-way process with in- and outflows happening at the same time. However, it is not the case of narrow and shallow central part of the Kerch Strait, which width and depth are equal to 3 km and 3-4 m. In absence of external forcing water exchange through the Kerch Strait indeed is a two-way process. However, due to the dominating role of a barotropic component in the total pressure gradient along the strait (Figure 5 in the revised version of the manuscript), even moderate wind forcing at the study region induces a one-way water exchange in the narrowest part of the strait which defines water exchange between the Azov and Black seas. Numerical simulations show that a two-way water exchange occurred in the Kerch Strait only during weak wind forcing conditions which total annual duration was $34 - 54$ days in $1992 - 2010$, which is only 9-15% of the whole year (Table 1). This result is also supported by previous in situ observations (Ivanov, 2011) and numerical modelling (Stanev et al., 2017) of water exchange in the Kerch Strait. Thus, one-way water exchange was observed during the majority of the year. This point was described in the text.

**3. As regards spreading of BP in the Sea of Azov using TSM satellite images, seems very unreliable information. As well as TSM was mentioned as not reliable tracer. Rather it can show wind-induced resuspension of sea-bottom sediments.**

Response:

TSM is not a reliable marker of SA water in the Black Sea because TSM increases over shallow areas in case of resuspension of bottom sediments. Thus, areas of elevated TSM can be induced by both sediment resuspension and spreading of SA water in the Black Sea. However, spreading of BS water in the Sea of Azov is indicated by reduced values of TSM in the shallow area adjacent to the Kerch Strait. Thus, resuspension of bottom sediments which increases TSM at the

study region can not be misleading for detection of SA water indicated by reduced TSM especially at shallow areas. Thus, TSM is not a reliable marker of SA water in the Black Sea, but is a reliable marker of BS water in the Sea of Azov. Moreover, in the revised version we analyzed both TSM and Chl-a and identified SA water in the Black Sea as areas of reduced TSM and Chl-a. This procedure provides more confidence in detection of SA water, as compared to usage of only TSM. Thus, the scheme of identification of BI events and detection of borders of BP at satellite imagery is relatively straightforward, as compared to identification of AI events. However, we assume that BP is not manifested by anomalies of TSM and Chl-a in the surface layer during low wind forcing conditions, while during strong wind forcing conditions resuspension of bottom sediments can induce elevated concentrations of TSM in the surface layer that also hinders identification of BP. As a result, many of BI events are not detected by optical satellite imagery. Also salinity distributions simulated by BSAS12 in the study region showed good agreement with detection BP in the Sea of Azov at satellite imagery, that is explicitly described in the revised version of the manuscript and illustrated by Figure 11.

**4. In my opinion in present stage, presented paper does not give correct or reliable information regarding water exchange between the Sea of Azov and Black Sea. However, current simulations by operative model or some additional measurements could increase value to this paper.**

Response:

According to your recommendation we applied calibrated hydrodynamic modelling using the BSAS12 model to prove the results obtained from the analysis of satellite imagery. Salinity distributions simulated by BSAS12 in the study region showed good agreement with detection of AP and BP at satellite imagery, that is explicitly described in the revised version of the manuscript and illustrated by Figures 2, 8, 11. Thus, the applied method of identification of AP and BP and the obtained results are supported by both satellite data and numerical modelling.

**5. As it is mentioned in the paper, circulation in the surface layer in the northeastern part of the Black Sea is dominated by a westward current along the continental slope and an anticyclonic eddy, which is regularly formed between this current and the coast near the Kerch Strait. How big influence eddy have on spreading the freshwater from the Sea of Azov.**

Response:

Thank you for this point. Mesoscale eddies formed in the Black Sea near the Kerch Strait can largely influence spreading and mixing pattern of SA waters that inflow to the Black Sea and propagate westward along the southeastern shore of the Crimean Peninsula. Satellite data and numerical modelling reveal that AP are regularly entrained by mesoscale eddies that can significantly intensify cross-shelf transport of freshened water inflowing from the Sea of Azov. The resulting transport of

freshened water to the deep part of the Black Sea can largely affect local biological and geochemical processes. This issue was discussed in the text, however, thorough study of this feature is beyond the scope of our study.

**6. On the page 6, author mentions that they use Chl-a as a stable tracer for AP in the Black Sea. Does this mean, that satellite images of Chl-a can be used only during big concentrations of Chl-a, and for example in winter periods this method cannot be used?**

Response:

Concentration of Chl-a in the Sea of Azov is larger than in the Black Sea during the whole year, including winter period. Thus, the applied method of detection of AP can be used for any periods of the year.

**References**

[revised manuscript text omitted]

~~As it was shown above, northeastern wind causes formation of AI event and prevents formation of BI event. BI event, in its turn, is formed in absence of northeastern wind and area of BP depends on duration of this period. This feature can be explained in the following way. Saline and dense water of the Black Sea tends to form a bottom advected gravity flow to the Sea of Azov through the Kerch Strait. In presence of northeastern wind forcing, wind stress is transferred into a relatively shallow Sea of Azov and induces intense flow of freshened water from the area adjacent to the Kerch Strait to the Black Sea. The Kerch Strait is very shallow, especially at its central part, which is 3-5 m deep. Thus, this wind induced surface flow occupies the whole water column in the Kerch Strait and prevents formation of gravity induced flow in the opposite direction in bottom layer. After the end of a period of northeastern wind forcing and cessation of AI, gravity flow from the Black Sea to the Sea of Azov is restored and BP is formed. As a result, area of BP depends on duration of the period without northeastern wind forcing.~~

[revised manuscript text omitted]

~~In this study we revealed wind forcing conditions that cause formation of AI and BI events and analyzed intra-annual variability of their monthly and annual frequency and duration for the period of 2000-2017. BI favorable wind conditions are significantly more frequent than AI favorable conditions and occur during 71% of days in a year with a distinct peak in May – July (85-89% of days). AI events slightly dominate over BI events only in October (58% of days), while during the rest of the year their share is 11-39%. Shifts between AI favorable and BI favorable wind conditions occur relatively rarely, 1-3 times in a month and only 46 times in a year. Thus, AI and BI events generally last for relatively long time periods; their mean annual durations are approximately 4.3 and 13.7 days, respectively. Duration of AI events does not significantly change during a year, while mean monthly duration of BI event varies from 5 to 30 days.~~

~~Also we obtained relations between spatial characteristics of AP, namely, alongshore scale and area, on wind forcing conditions during AI event. On the other hand, we did not determine relevant equations for water transport from the Black Sea to the Sea of Azov. It is caused by the fact that spatial characteristics and temporal variability of spreading of BP in the bottom layer of the Sea of Azov is much worse detected at satellite imagery, as compared to surface-advected AP. As a result, definition of dependencies for both discharge rate from the Black Sea to the Sea of Azov and spatial characteristics of BP on wind forcing requires specific in situ measurements which are beyond the current study.~~

[revised manuscript text omitted]

---

## Author Response (AR1)

**Reply to comments from the reviewers and summary of the major changes made in the revised manuscript OS-2018-2**

We strongly appreciate the reviewers' suggestions and comments that served to improve the article. In response to them,
5  first, we added results of numerical modelling that supported the results obtained from analysis of satellite imagery. Second, we clarified methods of detection of surface-advected outflows from the Sea of Azov to the Black Sea and bottom-advected inflows in the opposite direction based on agreement between Chl-a and TSM satellite imagery. Third, we modified the interpretation of influence of wing forcing on inflows from the Black Sea to the Sea of Azov and provided a more thorough physical background based on both satellite data and numerical modelling. Finally, we adopted many specific comments and
10  recommendations from the reviewers, reworked the structure of the manuscript, and improved the quality of English text and figures. The detailed reply to reviewers' comments and description of changes made in the manuscript are given below.

**Anonymous Referee  #1**

15  **The premise of this paper is a study the water exchange through the Kerch Strait. My assessment is that the authors rather studied sediment resuspension due to wind/wave events which is insufficient to infer the true water exchange that also exists without sediment addition. Sediment resuspension is highly variable, but this should not be used to claim that the water exchange is too. More specific ocean data with mooring measurements and/or calibrated hydrodynamic modelling would be better suited to quantify this water exchange. Without such, most of the paper**
20  **remains pure speculation and is of little scientific value.**

Thank you for this valuable comment. We totally agree that the sediment resuspension due to episodic wind/wave events strongly affects optical satellite imagery and can be misleading for identification of water exchange events. TSM distribution indeed increases over shallow areas in case of resuspension of bottom sediments. Thus, areas of elevated TSM can be
25  induced by both sediment resuspension and spreading of AP. However, in the revised version of the manuscript we used joint analysis of sea surface temperature (SST), concentrations of chlorophyll a (Chl-a) and total suspended matter (TSM) retrieved from optical satellite data. We applied the following scheme of identification of AI events and detection of borders of AP basing on satellite data. Inflow events were identified by elevated concentration of Chl-a in the Kerch Strait and the adjacent coastal area of the Black Sea, because Chl-a has the lowest short-term variability among the considered sea surface
30  characteristics and are not affected by wind/wave-induced sediment resuspension. If an inflow event was detected, we analyzed areas of elevated TSM, Chl-a, and elevated (in summer) or reduced (in winter) SST associated with formation of AP in the northeastern part of the Black Sea. If general forms and spatial scales of these areas were similar, we defined borders of AP basing on gradient of TSM that is the most stable passive tracer of AP in absence of episodic wind-induced

bottom resuspension events. If areas of TSM, Chl-a, and SST anomalies were not consistent with each other, we assumed that areas of elevated TSM and reduced SST (in winter) were modified by wind-induced resuspension and mixing of AP with subjacent sea. In this case we defined borders of AP basing on gradient of Chl-a that is the most stable tracer of AP during intense wind forcing conditions. These issues were described in the text. Also according to your recommendation we applied a calibrated hydrodynamic modelling using the BSAS12 model to prove the results obtained from the analysis of satellite imagery. Salinity distributions simulated by BSAS12 in the study region showed good agreement with detection of AP and BP at satellite imagery, that is explicitly described in the revised version of the manuscript and illustrated by Figures 2, 8, 11. Thus, the applied method of identification of AP and BP and the obtained results are supported by both satellite data and numerical modelling.

**A starting point should be to explore the seasonal cycle of density differences across the strait to identify any possible seasonality of the forcing. The existence of a density gradient (in conjunction with a barotropic pressure gradient) should induce a steady exchange circulation typical for positive estuaries.**

Thank you for this comment. Based on the results of numerical modelling, we explored variability of baroclinic and barotropic force in the Kerch Strait during 1992 – 2017. We show that the total pressure gradient along the strait is governed by a barotropic component, due to relatively large average difference of water level (0.1 m) in the southern and northern ends of the strait. Density gradient along the strait indeed exists; however, its value does not exceed 6 kg/m$^3$. As a result, the role of a barotropic component in the total pressure gradient in the strait is by one order of magnitude greater than the role of a baroclinic component. Thus, circulation through the Kerch Strait is not steady and unidirectional, but has large synoptic variability of intensity and direction governed by episodic wind forcing events. Also annual variability of the total pressure gradient in the strait does not show any seasonality. It supports our results that water exchange through the Kerch Strait has wind-govern synoptic variability, and do not show seasonal dependence on river discharge rate to the Sea of Azov. These issues were thoroughly described and discussed in the manuscript.

**Indeed the water exchange through the mouth of a positive estuary is typically a two-way process with in- and outflows happening at the same time. Indeed, this exchange may be weaker or enhanced by winds as demonstrated in several studies before. To claim that winds are the sole initiator of episodic exchange flows is not aligned with previous studies on density-driven exchange flows in positive estuaries.**

Many thanks for this important comment. We totally agree that water exchange through the mouth of a positive estuary is typically a two-way process with in- and outflows happening at the same time. However, it is not the case of narrow and shallow central part of the Kerch Strait, which width and depth are equal to 3 km and 3-4 m. In absence of external forcing water exchange through the Kerch Strait indeed is a two-way process. However, due to the dominating role of a barotropic

component in the total pressure gradient along the strait, even moderate wind forcing at the study region induces a one-way water exchange in the narrowest part of the strait which defines water exchange between the Azov and Black seas. Numerical simulations show that a two-way water exchange occurred in the Kerch Strait only during weak wind forcing conditions which total annual duration was 34 – 54 days in 1992 – 2010, which is only 9-15% of the whole year (Table 1). Thus, one-way water exchange was observed during the majority of the year. This result is also supported by previous in situ observations (Ivanov, 2011) and numerical modelling (Stanev et al., 2017) of water exchange in the Kerch Strait. This point was described in the text.

**Section 2: Give information on seasonal variations of density and differences across the Strait.**

We agree with this point. We added variability of sea level and density gradients, as well as total pressure gradient along the Kerch Strait during 1997 and 2007 obtained from numerical modelling (Figure 5).

**Page 4, line 21: A persistent residual tidal flow of ~5 cm/s could move substantial volumes of water through the strait as compared with rare episodes.**

Barotropic tidal current in the Kerch Strait is less than 5 cm s$^{-1}$ except the narrowest part of the strait where they are equal to 6-10 cm s$^{-1}$ during peak flow (Ferrain et al., 2018). However, maximal velocity of 2-days averaged tidal current in the Kerch Strait is less than 5 cm s$^{-1}$ and its flow direction reverses during the tidal cycle. Thus, tidal current does not form a persistent residual flow and its role in water exchange between the Black and Azov seas can be regarded as negligible. This issue was clarified in the text.

**Page 7: The authors claim that the area of an AP can substantially increase over time. In fact, it's the effect of mesoscale eddies (as obvious from the images) and the associated entrainment that causes this effect. However, this feature does NOT imply any increase in freshwater discharge rate.**

Many thanks for this comment. We agree that increase of the area of AP showed at the related figure is caused by entrainment by a mesoscale eddy that was proved by numerical modelling. Thus, the observed feature does not relate to increase in freshwater transport through the Kerch Strait. In the revised version of the manuscript we verified all cases of freshwater inflow detected at satellite imagery by numerical modelling. We selected only AP that were spreading as an alongshore current and did not consider AP were entrained by a mesoscale eddy. Thus, in the revised version of the manuscript we modified dependences between wind forcing conditions and spatial/volume characteristics of AP which reproduce intensity of freshwater discharge through the Kerch Strait in response to local wind forcing conditions.

**Anonymous Referee #2**

In this paper, the authors consider the Sea of Azov (SA) as an estuary and investigate a flow exchange between SA and the adjacent Black Sea (BS) by means of remote sensing. As a proxy for the buoyant outflow from SA into BS they use Cl-a, although on page 5 (lines 22-25) they state that SST, Cl-a and TSM "are prone to significant variability and/or do not act as passive tracers, which hinders their direct application for accurate identification" of buoyant outflow from SA.

Thank you for this valuable comment. SST, Chl-a, and TSM indeed are prone to significant variability and straightforward
10 usage of these individual characteristics for detection of outflow from SA to BS can be misleading. Thus, in the revised version of the manuscript we used more comprehensive joint analysis of SST, Chl-a, and TSM. Inflow events from SA to BS were identified by elevated concentration of Chl-a in the Kerch Strait and the adjacent coastal area of the Black Sea, because Chl-a has the lowest short-term variability among the considered sea surface characteristics and are not affected by wind/wave-induced sediment resuspension. If an inflow event was detected, we analyzed areas of elevated TSM, Chl-a, and
15 elevated (in summer) or reduced (in winter) SST associated with formation of the Azov plume (AP) in the northeastern part of the Black Sea. If general forms and spatial scales of these areas were similar, we defined borders of AP basing on gradient of TSM that is the most stable passive tracer of AP in absence of episodic wind-induced bottom resuspension events. If areas of TSM, Chl-a, and SST anomalies were not consistent with each other, we assumed that areas of elevated TSM and reduced SST (in winter) were modified by wind-induced resuspension and mixing of AP with subjacent sea. In this case we defined
20 borders of AP basing on gradient of Chl-a that is the most stable tracer of AP during intense wind forcing conditions. These issues were described in the text. Also we applied a calibrated hydrodynamic modelling using the BSAS12 model to prove the results obtained from the analysis of satellite imagery. Salinity distributions simulated by BSAS12 in the study region showed good agreement with detection of AP and BP at satellite imagery, that is explicitly described in the revised version of the manuscript and illustrated by Figures 2, 8, 11. Thus, the applied method of identification of AP and BP and the
25 obtained results are supported by both satellite data and numerical modelling.

As a proxy for the BS water inflow into SA, the authors use TSM, which they have discarded as a reliable marker for buoyant water just one page earlier (i.e., page 7, lines 5-8). They have to rely on TSM signal now, because the BS inflow propagates near the bottom and does not produce immediate signature on the surface.
30 Frankly, I don't think that TSM signal can provide any reliable information about the presence of BS water at the bottom, it just tells us about the wind-induced resuspension of sediments.

TSM is not a reliable marker of SA water in the Black Sea because TSM increases over shallow areas in case of resuspension of bottom sediments. Thus, areas of elevated TSM can be induced by both sediment resuspension and

spreading of SA water in the Black Sea. However, spreading of BS water in the Sea of Azov is indicated by reduced values of TSM in the shallow area adjacent to the Kerch Strait. Thus, resuspension of bottom sediments which increases TSM at the study region can not be misleading for detection of SA water indicated by reduced TSM especially at shallow areas. Thus, TSM is not a reliable marker of SA water in the Black Sea, but is a reliable marker of BS water in the Sea of Azov.

5 Moreover, in the revised version we analyzed both TSM and Chl-a, and identified SA water in the Black Sea as areas of reduced TSM and Chl-a. This procedure provides more confidence in detection of SA water, as compared to usage of only TSM. As a result, the scheme of identification of inflow events from the Black Sea to the Sea of Azov and detection of borders of the bottom-advected Black Sea plume (BP) at satellite imagery is relatively straightforward, as compared to identification of inflow events from the Sea of Azov to the Black Sea. However, we assume that BP is not manifested by

10 anomalies of TSM and Chl-a in the surface layer during low wind forcing conditions, while during strong wind forcing conditions resuspension of bottom sediments can induce elevated concentrations of TSM in the surface layer that also hinders identification of BP. As a result, many of BI events are not detected by optical satellite imagery.

**The major conclusion of this study is that the buoyant outflow from SA into BS occurs only under the external**
15 **forcing by northeasterly (NE) winds, while relaxation of NE winds occurs during the relaxation of NE winds (e.g., page 10, line 24) and under any wind conditions. The authors also attempt to quantify "intensity" of the flow exchange through the Kerch Strait by determining the area of the SA plume in BS, and scaling this area against the integral of wind speed when the wind is favorable for the plume formation. In my opinion, there is a major confusion in this conclusion: do the authors imply that NE winds precondition the BS inflow into SA (that is, relaxation of NE**
20 **winds triggers the BS inflow) or do they mean that the BS inflow occurs always as long as NE winds do not operate? The authors seem to interchange these two very different messages in different parts of their manuscript (e.g., compare lines 24-26 on page 10 and lines 23-26 on page 11).**

Many thanks for this important comment. In the revised version of the manuscript we clarified meaning of "relaxation of NE
25 winds" related to inflow of BS waters into SA. We state that inflow of BS waters into SA occurs after reverse of a barotropic pressure gradient as a result of relaxation of NE winds, i.e. relaxation of NE winds triggers the BS inflow. This result was supported by numerical modelling.

**I feel strongly that the estuarine exchange flow should occur in both directions, but the authors seem to imply that**
30 **under light or no wind forcing conditions, it is a one-way traffic: BS water flows in SA in a unidirectional manner. This is a bold and unsubstantiated claim.**

Many thanks for this important comment. We totally agree that water exchange through the mouth of a positive estuary is typically a two-way process with in- and outflows happening at the same time. However, it is not the case of narrow and

shallow central part of the Kerch Strait, which width and depth are equal to 3 km and 3-4 m. In absence of external forcing water exchange through the Kerch Strait indeed is a two-way process. However, due to the dominating role of a barotropic component in the total pressure gradient along the strait (Figure 5 in the revised version of the manuscript), even moderate wind forcing at the study region induces a one-way water exchange in the narrowest part of the strait which defines water exchange between the Azov and Black seas. Numerical simulations show that a two-way water exchange occurred in the Kerch Strait only during weak wind forcing conditions which total annual duration was 34 – 54 days in 1992 – 2010, which is only 9-15% of the whole year (Table 1). Thus, one-way water exchange was observed during the majority of the year. This result is also supported by previous in situ observations (Ivanov, 2011) and numerical modelling (Stanev et al., 2017) of water exchange in the Kerch Strait. The rest of the year a one-way water exchange was observed. This point was described in the text.

**Apart from this major issue with paper's conclusions, I have some questions with the analysis. I am not sure why the authors use logarithms of properties compared in Figure 6 (and not the properties themselves). The bottom panel scatterplot does not resemble a linear function, so the linear regression is a poor choice here. This relationship is by no means linear: first, the wind driven transport is proportional to the wind stress (which is quadratic in wind speed). Second, the surface area is not directly proportional to the discharge Q through the Kerch Strait, but rather to Q/h, where h is the sickness of the plume, which is generally unknown but does depend on the wind stress.**

We agree that scatterplots in this figure (Figure 9 in the revised version of the manuscript) do not resemble linear functions, so the linear regression is a poor choice. In the revised version of the manuscript we improved these dependences and their approximations. Alongshore extent and area of AP increase with increase of wind forcing index. However, this increase is not steady, its derivative decreases with increase of wind forcing index. In particular if wind forcing index exceeds 3500 km, alongshore extent and area of AP are almost stable. Thus, the observed forms of dependences between have good approximation by logarithmic functions. We are not aware that logarithmic approximation has any straightforward physical background (despite steady decrease of dependence derivative) due to complex and strongly non-linear dependences between wind speed, discharge through the Kerch Strait, and spatial characteristics of AP. However, the obtained relations are essential for numerical parameterizations of water exchange through the Kerch Strait based on wind data, which are addressed in this study.

**Regarding the upper panel of Figure 6, I think the alongshore extension of the plume under the downwelling wind forcing primarily depends on the ALONG-SHELF wind stress component (not the wind velocity magnitude).**

We agree with this point. Spatial scales of a buoyant plume in a non-tidal sea are defined mainly by two external forcing parameters, namely, river discharge rate and wind forcing. Thus, alongshore extent of AP depends on discharge rate of SA

water through the Kerch Strait and local wind forcing. Inflow of water from SA to BS and formation of AP occurs only under north-easterly wind forcing. As a result, cross-shelf wind component determines discharge rate of SA water through the Kerch Strait and both wind components govern further alongshore extension of AP. In this study, we consider alongshore extension and area of AP only during periods of inflow from SA to BS, i.e., we do not consider AP detached from its source in the Kerch Strait after reverse of flow direction in the strait and cessation of inflow from SA to BS. Thus, we calculate alongshore extension and area of AP only during moderate and strong north-easterly winds that induce inflow from SA to BS. During these periods, alongshore extension of AP indeed shows good relation with variability both cross- shelf and along-shelf wind component. However, the dependence of alongshore extension of AP on wind velocity magnitude is better due to small range of the considered wind directions. This issue was clarified in the text.

**Throughout the paper, there is unfortunate confusion about the wind direction. In geosciences, wind direction indicates where the wind blows from and it is measured from true north in clockwise direction. Wind direction 210 to 260 degrees (page 6, line 17) means that the wind blows from southwest, and it's not what the authors assume. Likewise, "northeastern" (wind) and "northeastward" have opposite meanings, northeastward denotes a flow from southwest towards northeast, but the authors freely interchange these two terms (e.g., lines 24 and 26 on page 10, and many other instances).**

Thank you for this point, we checked all parts of the manuscript where wind direction was mentioned and corrected the wrong usage of the related words.

**On page 11 (2nd paragraph), the authors compare the SA volume and the annual volume of the freshwater discharge. This is a very strange proposition because a year is not a proper time scale for the exchange processes between AS and BS, according to the authors. In my opinion, the authors should rectify their arguments in this part of the discussion.**

We agree with the reviewer that we should improve arguments in this part of the discussion. In this paragraph we discuss relation between freshwater discharge to river estuary and water exchange between the estuary and open sea, i.e. does signal of river discharge dissipate in the estuary or it influences freshwater outflow from estuary to open sea. This process depends on two main factors, namely, volume of inflowing river discharge and spatial scales of an estuary. If large river inflows to small estuary (e.g., the Amur River and the Amur Liman), variability of river discharge rate strongly affects water outflow from estuary to open sea. On the other hand, discharge of a small river inflowing to a large estuary does not influence and water exchange between the estuary and open sea. The area and volume of the Sea of Azov are large enough; therefore, signals of synoptic and even significant seasonal variability of discharge of the DON and Kuban rivers dissipate in the Sea of Azov and do not influence formation and/or intensity of AI events. It was clarified in the manuscript

Finally, the language should be checked throughout. For instance, it is a "boundary condition", not a "border condition". Word "significant" is badly overused. In scientific literature, this word tends to have a statistical context (that is, significant in quantitative, statistical sense). "Substantial" would probably sound better and less irritating for quantitatively minded readers.

Thank you for these points, we adopted them.

**Anonymous Referee #3**

The purpose of this paper was to analyze water exchange between Sea of Azov and Black sea. For this analyze author used satellite images and wind data. The author use TSM and CHL-a characteristics to trace spreading of AP and BP in the area. However on the page 5, author claims that: "above mentioned characteristics are prone to significant variability and not act as passive tracers".

Thank you for this valuable comment. In the revised version of the manuscript we used joint analysis of sea surface temperature (SST), concentrations of chlorophyll a (Chl-a) and total suspended matter (TSM) retrieved from optical satellite data. These characteristics are prone to significant variability and not act as passive tracers. Thus these characteristics cannot be used for straightforward identification of inflow of freshened waters from the Sea of Azov to the Black Sea. However, these processes and their temporal scales are different for SST (diurnal cycle of solar radiation), TSM (episodic wind-induced bottom resuspension events), and Chl-a (synoptic and seasonal biological cycles). Thus, joint analysis of SST, TSM, and Chl-a distributions can be used for accurate detection of spreading of AP in the Black Sea. We applied the following scheme of identification of AI events and detection of borders of AP basing on satellite data. Inflow events were identified by elevated concentration of Chl-a in the Kerch Strait and the adjacent coastal area of the Black Sea, because Chl-a has the lowest short-term variability among the considered sea surface characteristics and are not affected by wind/wave-induced sediment resuspension. If an inflow event was detected, we analyzed areas of elevated TSM, Chl-a, and elevated (in summer) or reduced (in winter) SST associated with formation of AP in the northeastern part of the Black Sea. If general forms and spatial scales of these areas were similar, we defined borders of AP basing on gradient of TSM that is the most stable passive tracer of AP in absence of episodic wind-induced bottom resuspension events. If areas of TSM, Chl-a, and SST anomalies were not consistent with each other, we assumed that areas of elevated TSM and reduced SST (in winter) were modified by wind-induced resuspension and mixing of AP with subjacent sea. In this case we defined borders of AP basing on gradient of Chl-a that is the most stable tracer of AP during intense wind forcing conditions. These issues were described in the text. Also we applied calibrated hydrodynamic modelling using the BSAS12 model to prove the results

obtained from the analysis of satellite imagery. Salinity distributions simulated by BSAS12 in the study region showed good agreement with detection of AP and BP at satellite imagery, that is explicitly described in the revised version of the manuscript and illustrated by Figures 2, 8, 11. Thus, the applied method of identification of AP and BP and the obtained results are supported by both satellite data and numerical modelling.

**According to this paper AI events are induced by northeastern wind events, with speed >5 m/s, and during all other wind conditions the unidirectional flow towards the north occur. I would agree that during some exceptional events unidirectional flow can take place, but would suggest that water exchange in estuary should be two-way process, with simultaneous outflow and inflow events.**

Many thanks for this important comment. We totally agree that water exchange through the mouth of a positive estuary is typically a two-way process with in- and outflows happening at the same time. However, it is not the case of narrow and shallow central part of the Kerch Strait, which width and depth are equal to 3 km and 3-4 m. In absence of external forcing water exchange through the Kerch Strait indeed is a two-way process. However, due to the dominating role of a barotropic

15 component in the total pressure gradient along the strait (Figure 5 in the revised version of the manuscript), even moderate wind forcing at the study region induces a one-way water exchange in the narrowest part of the strait which defines water exchange between the Azov and Black seas. Numerical simulations show that a two-way water exchange occurred in the Kerch Strait only during weak wind forcing conditions which total annual duration was 34 − 54 days in 1992 − 2010, which is only 9-15% of the whole year (Table 1). This result is also supported by previous in situ observations (Ivanov, 2011) and

20 numerical modelling (Stanev et al., 2017) of water exchange in the Kerch Strait. Thus, one-way water exchange was observed during the majority of the year. This point was described in the text.

**As regards spreading of BP in the Sea of Azov using TSM satellite images, seems very unreliable information. As well as TSM was mentioned as not reliable tracer. Rather it can show wind-induced resuspension of sea-bottom sediments.**

TSM is not a reliable marker of SA water in the Black Sea because TSM increases over shallow areas in case of resuspension of bottom sediments. Thus, areas of elevated TSM can be induced by both sediment resuspension and spreading of SA water in the Black Sea. However, spreading of BS water in the Sea of Azov is indicated by reduced values of TSM in the shallow area adjacent to the Kerch Strait. Thus, resuspension of bottom sediments which increases TSM at the

30 study region can not be misleading for detection of SA water indicated by reduced TSM especially at shallow areas. Thus, TSM is not a reliable marker of SA water in the Black Sea, but is a reliable marker of BS water in the Sea of Azov. Moreover, in the revised version we analyzed both TSM and Chl-a and identified SA water in the Black Sea as areas of reduced TSM and Chl-a. This procedure provides more confidence in detection of SA water, as compared to usage of only TSM. Thus, the scheme of identification of BI events and detection of borders of BP at satellite imagery is relatively

straightforward, as compared to identification of AI events. However, we assume that BP is not manifested by anomalies of TSM and Chl-a in the surface layer during low wind forcing conditions, while during strong wind forcing conditions resuspension of bottom sediments can induce elevated concentrations of TSM in the surface layer that also hinders identification of BP. As a result, many of BI events are not detected by optical satellite imagery. Also salinity distributions

5    simulated by BSAS12 in the study region showed good agreement with detection BP in the Sea of Azov at satellite imagery, that is explicitly described in the revised version of the manuscript and illustrated by Figure 11.

**In my opinion in present stage, presented paper does not give correct or reliable information regarding water exchange between the Sea of Azov and Black Sea. However, current simulations by operative model or some**

10    **additional measurements could increase value to this paper.**

According to your recommendation we applied calibrated hydrodynamic modelling using the BSAS12 model to prove the results obtained from the analysis of satellite imagery. Salinity distributions simulated by BSAS12 in the study region showed good agreement with detection of AP and BP at satellite imagery, that is explicitly described in the revised version

15    of the manuscript and illustrated by Figures 2, 8, 11. Thus, the applied method of identification of AP and BP and the obtained results are supported by both satellite data and numerical modelling.

**As it is mentioned in the paper, circulation in the surface layer in the northeastern part of the Black Sea is dominated by a westward current along the continental slope and an anticyclonic eddy, which is regularly formed between this**

20    **current and the coast near the Kerch Strait. How big influence eddy have on spreading the freshwater from the Sea of Azov.**

Thank you for this point. Mesoscale eddies formed in the Black Sea near the Kerch Strait can largely influence spreading and mixing pattern of SA waters that inflow to the Black Sea and propagate westward along the southeastern shore of the

25    Crimean Peninsula. Satellite data and numerical modelling reveal that AP are regularly entrained by mesoscale eddies that can significantly intensify cross-shelf transport of freshened water inflowing from the Sea of Azov. The resulting transport of freshened water to the deep part of the Black Sea can largely affect local biological and geochemical processes. This issue was discussed in the text, however, thorough study of this feature is beyond the scope of our study.

30    **On the page 6, author mentions that they use Chl-a as a stable tracer for AP in the Black Sea. Does this mean, that satellite images of Chl-a can be used only during big concentrations of Chl-a, and for example in winter periods this method cannot be used?**

Concentration of Chl-a in the Sea of Azov is larger than in the Black Sea during the whole year, including winter period. Thus, the applied method of detection of AP can be used for any periods of the year.

**6 Conclusions**

In this work we studied water exchange between the Sea of Azov to the Black Sea through the Kerch Strait. We revealed that different physical mechanisms govern water transport in southward (from the Sea of Azov to the Black Sea, AI events) and northward (from the Black Sea to the Sea of Azov, BI events) directions. Analysis of satellite imagery, wind data, and numerical model outputs shows that water exchange in the Kerch Strait is governed by a wind-induced barotropic pressure gradient. As a result, water flow through the shallow and narrow Kerch Strait is a one-way process during the majority of the time. Southward AI events are induced by moderate and strong northeasternnorth-easterly wind forcing. In this case, wind stress causes transport of freshened water from shallow southern part of the Sea of Azov through the Kerch Strait that, which results in formation of a buoyant plume in the Black Sea. This surface-advected Azov Sea water plume (AP) is characterized by elevated concentrations of suspended sediments and chlorophyll a and is distinctly visiblecan be detected at optical satellite imagery. AP is spreading off the Kerch Strait as a quasi-geostrophic coastal current along the southeastern shore of the Crimean Peninsula, its area steadily increases during an AI event that last 1-5 days. As a result, AP occupies a large area in the northeastern part of the Black Sea up to 2000 km². However, AP dissipates during 1-5 days after end of AI event. The short-term, but regular process of formation and spreading of AP at the northeastern part of the Black Sea influences many local physical, biochemical, and geological processes, which were addressed in many previous studies.

Northward water transport in the Kerch Strait, on the opposite, was registered after relaxation of strong northeastwardnorth-easterly winds. Saline that results in reverse of the barotropic pressure gradient along the strait and dense watertriggers inflow from the Black Sea propagates through the Kerch Strait and inflows to the freshened Sea of Azov as a bottom-advected flow that is induced by the gravity force. Strong northeastern. Thus, strong north-easterly wind plays a restricting role for this process, because intense wind-induced southward surface flow of water from the Sea of Azov occupies the whole water column in the shallow centralnorthern part of the Kerch Strait. This prevents formation of the gravity-induced flow of water from the Black Sea in the opposite direction in the bottom layer. Analysis of satellite images did not show any direct dependence of northward water transport in the Kerch Strait on characteristics of local wind forcing. However, this feature can be caused by relatively low number of detected BI events atin satellite imagery, therefore future study of influence. Future studies of wind forcing influence on this process will thus require more specific and detailed in situ measurements and/or numerical modelling in the study region.

After water from the Black Sea inflows to the Sea of Azov it forms a bottom advected BP that tends to spread down the slope due to gravity force and accumulate in the deep central basin of the Sea of Azov. Area of BP depends on duration of the related BI event, i.e., period without northeastern wind forcing that can last up to 13 days. BP exhibits intense mixing

We determined that wind forcing governs direction and intensity of water transport in the Kerch Strait on inter-annual time scale. River runoff to the Sea of Azov does not show any distinct influence on synoptic and seasonal variability of water exchange through the Kerch Strait , i.e., signal of river discharge dissipates in the Sea of Azov and does not influence freshwater outflow from estuary to open sea. This feature is not typical for large river estuaries,  e.g., the Patos Lagoon (Castelao and Moller, 2006) and the Amur Liman (Osadchiev, 2017). Relation between freshwater discharge to river estuary and water exchange between the estuary and open sea depends on two main factors, namely, the volume of inflowing river discharge and the spatial scales of an estuary. The volume of the Sea of Azov (290 km$^3$) is greater by one order of magnitude than the annual continental discharge to the sea (20 – 54 km$^3$). River runoff during flooding period increases the level of the Sea of Azov only by 6-7 centimetres as compared to draught period. As a result, signal of seasonal discharge variability of the Don and Kuban rivers dissipates in the Sea of Azov and does not influence the intensity of water exchange through the Kerch Strait. The volume of the Patos Lagoon (50 km$^3$), on the opposite, is of the same order as continental runoff volume (75 km$^3$), which causes an increase of the lagoon level by 70-80 cm during freshet periods. As a result, stable seaward flow from the Patos Lagoon during the seasonal flood can be reversed only by very strong winds (Moller and Castaing, 1999). The volume of the Amur Liman (20 km$^3$) is much less than the annual Amur discharge volume (400 km$^3$). Therefore, water exchange between the Amur Liman and sea is dominated by the river regime during the major part of a year (Osadchiev, 2017).

~~In this study we revealed wind forcing conditions that cause formation of AI and BI events and analyzed intra-annual variability of their monthly and annual frequency and duration for the period of 2000-2017. BI favorable wind conditions are significantly more frequent than AI favorable conditions and occur during 71% of days in a year with a distinct peak in May – July (85-89% of days). AI events slightly dominate over BI events only in October (58% of days), while during the rest of the year their share is 11-39%. Shifts between AI favorable and BI favorable wind conditions occur relatively rarely, 1-3 times in a month and only 46 times in a year. Thus, AI and BI events generally last for relatively long time periods; their mean annual durations are approximately 4.3 and 13.7 days, respectively. Duration of AI events does not significantly change during a year, while mean monthly duration of BI event varies from 5 to 30 days.~~

~~Also we obtained relations between spatial characteristics of AP, namely, alongshore scale and area, on wind forcing conditions during AI event. On the other hand, we did not determine relevant equations for water transport from the Black Sea to the Sea of Azov. It is caused by the fact that spatial characteristics and temporal variability of spreading of BP in the bottom layer of the Sea of Azov is much worse detected at satellite imagery, as compared to surface-advected AP. As a result, definition of dependencies for both discharge rate from the Black Sea to the Sea of Azov and spatial characteristics of BP on wind forcing requires specific in situ measurements which are beyond the current study.~~

[revised manuscript text omitted]

---

## Referee Report (RR1)

Review
of a paper by Ivan Zavialov, Alexander Osadchiev, Roman Sedakov, Bernard Barnier, Jean-Marc Molines, Vladimir Belokopytov
"Water exchange between the Sea of Azov and the Black Sea through the Kerch Strait"

The paper is devoted to the description of the mechanisms forcing the penetration of low-saline waters from the Sea of Azov to the northeastern part of the Black Sea (and their subsequent distribution in the Black Sea) and high-saline Black Sea waters to the Azov Sea through the narrow Kerch Strait. Based on the analysis of satellite data and numerical simulation results, it is convincingly shown that the main driving force of this penetration is the action of the northeast wind. Water flow through the shallow and narrow Kerch Strait is a one-way process during the majority of the time. However the penetration of the Black Sea waters into the Sea of Azov is observed after the end of a long action of a strong northeastern wind. In both cases, the barotropic pressure gradient along the strait plays the primary role in the water exchange between the seas. The variations in freshwater runoff do not significantly affect the water exchange on an intra-annual time scale. It is also clearly shown that the Azov Sea waters in the Black Sea most often spread along the coast of the Crimea peninsula as a buoyant plume. The area of this plume and the distance to which it spreads along the coast of Crimea are functions of the product of the average northeast wind speed and the duration of its action.

The paper is very interesting and contains new significant scientific results, some of which are listed above. It should be published in the OS Journal after eliminating some of the shortcomings noted below.

1. In Section 2 «Study Area», too much attention is paid to the physical-geographical description of the Sea of Azov, which is not related to the main objectives of the paper. This part is proposed to be reduced somewhat.
2. Section 5 "Discussion and Conclusions" does not consider the impact of the Black Sea dynamics on the AP propagation distance along the Crimean Peninsula. However, in case with the developed Rim current, the most distant propagation of AP along the coast can be expected, and in case with the presence of a mesoscale anticyclonic eddy near the Kerch strait - the least distant propagation of AP can be expected. This issue should be considered.
3. On page 8 it is indicated that "stable density gradient that exists along the strait doesn't exceed 6 kg/m$^3$". An error is made here, since the indicated value characterizes not the density gradient along the strait, but the density difference, or jump along it.
4. On page 10, the "wind forcing index" is introduced, which is the product of the average speed of the northeast wind and the duration of its action. The authors are looking for the dependences of the AP area in the Black Sea and the AP propagation distance along the coast of Crimea from this parameter. However, the index, which is the product of the wind stress and its duration, should have a more clear physical value, because the wind stress (not its speed!) determines the Ekman transport, which creates a barotropic pressure gradient along the strait. Authors should find the dependencies of the above mentioned characteristics of the AP on this parameter.
5. The caption under fig. 5 indicates that the various graphs show the gradients of different characteristics along the Kerch Strait. In fact, they are not gradients but jumps, or differences along the strait.

---

## Author Response (AR2)

**Author's reply to comments from the reviewers and summary of the major changes made in revised manuscript OS-2019-2**

We appreciate the reviewers' suggestions and comments that served to improve the article. In response to them we clarified
5  details about model setting and model validation, reworked numerical dependencies between wind forcing and AP spatial characteristics, and significantly improved the quality of English text and figures. The detailed reply to reviewers' comments and description of changes made in the manuscript are given below.

**Reviewer #1**

*In Section 2 «Study Area», too much attention is paid to the physical-geographical description of the Sea of Azov, which is not related to the main objectives of the paper. This part is proposed to be reduced somewhat.*

Thank you for this comment. In this paper we indeed provided a detailed geographical description of the study region. We significantly shortened this part of the manuscript.

*Section 5 "Discussion and Conclusions" does not consider the impact of the Black Sea dynamics on the AP propagation distance along the Crimean Peninsula. However, in case with the developed Rim current, the most distant propagation of AP along the coast can be expected, and in case with the presence of a mesoscale anticyclonic eddy near the Kerch strait - the least distant propagation of AP can be expected. This issue should be considered.*

20  Thank you for this important comment. We totally agree that dynamics of the Black Sea influences spreading patterns of AP. We added discussion of this issue to Section 4.2.

*On page 8 it is indicated that "stable density gradient that exists along the strait doesn't exceed 6 kg/m3". An error is made here, since the indicated value characterizes not the density gradient along the strait, but the density difference, or jump*
25  *along it.*
Thank you for this point. Corrected.

*On page 10, the "wind forcing index" is introduced, which is the product of the average speed of the northeast wind and the duration of its action. The authors are looking for the dependences of the AP area in the Black Sea and the AP propagation*
30  *distance along the coast of Crimea from this parameter. However, the index, which is the product of the wind stress and its duration, should have a more clear physical value, because the wind stress (not its speed!) determines the Ekman transport, which creates a barotropic pressure gradient along the strait. Authors should find the dependencies of the above mentioned characteristics of the AP on this parameter.*

Many thanks for this important comment. We agree that numerical dependencies between wind forcing and AP spatial characteristics should be based on wind stress rather than wind speed. Thus, we reworked the obtained dependences described in Section 5.

*The caption under fig. 5 indicates that the various graphs show the gradients of different characteristics along the Kerch Strait. In fact, they are not gradients but jumps, or differences along the strait.*

Thank you for this point. Corrected.

10 **Reviewer #2**

*Although the latest version of the manuscript has considerably improved, major revisions are still necessary. It was a very good idea to add the results of a numerical circulation model to the analysis of satellite images. However, the horizontal resolution with 6.75 km is rather coarse because I assume that the baroclinic Rossby radius is very often below 20 km. In*

15 *addition, the Kerch Strait is only 3 km narrow prohibiting a proper representation of the exchange flow through the strait. Further, a reference introducing this new model setup is not provided. Hence, I would like to see evaluation figures at least in the attachment, e.g. mean temperature and salinity profiles for 1992-2017.*

Thank you for this important comment. We totally agree that the model setting and model validation should be described in more detail. The BSAS12 model is a new configuration of NEMO (version 3.6) and the current paper is the first one using

20 simulations carried out with this model. BSAS12 uses the latest NEMO advances in parameterisations and numerical schemes. In particular, it uses a momentum advection scheme that has an implicit scale-selective diffusion term that depends on the gradients of the instantaneous velocity field (the Upstream-Biased Scheme, UBS, Shchepetkin and McWilliams, 2005, see also the NEMO documentation). This scheme prevents the use of an explicit viscosity and significantly reduces the dissipation of the mesoscale instabilities. We provide a short assessment of the modelled surface circulation in the

25 Supplementary Material. Currents, eddy kinetic energy, T-S census are shown and discussed for the whole Black Sea basin as well for the Kerch Strait regon.

The theoretical eddy-scale can be estimated to be one-half of the length-scale associated to the first radius of deformation. A deformation radius of ~20 km in the open sea implies an eddy-scale of ~60 km (see Supplementary Material), so the model has 9 grid-points per half-wavelength, which is enough to resolve the largest eddies. Nevertheless, we consider this model as

30 eddy-permitting (not eddy-resolving) because it only resolves the large eddies, and does not properly resolve the shelf eddies that have a smaller characteristic length-scale. As shown in the Supplementary Material, the model generates mesoscale eddies of realistic amplitude in the open sea (the Sevastopol eddy, the Batumi, and much smaller eddy features on the shallow shelves.

The Kerch Strait has been widened to be represented with a grid-resolution of 6.75 km (see Supplementary Material) and friction parameters in the strait were tuned to produce a transport through the strait that is in a reasonable agreement with estimates published in the literature, as discussed in the paper. Note that if the transport through the strait is consistent with hydraulic control, it will depend to first order to the pressure gradient (i.e. wind stress) on each side of the strait rather than

5    from the details of the flow within the strait. This is very likely why the model, although its representation of the geometry of the Kerch Strait is somewhat different from the real one, does produce realistic estimates of the transport.

*Please improve the language. A few examples are listed below. The usage of articles should be checked by a native English speaking person.*

10    The text was proofread and corrected by an expert English speaker.

*Minor comments:*
*P1 L11 "flow into" instead of "inflow!*
*P1 L20 "wind-governed"*
15    *P2 L3 "with the more saline Black Sea"*
*P2 L22 "on the coastal ecosystem"*
*P2 L27 "the water exchange"*
*P2 L28 "and the Black Sea"*
*P1 L10, P2 L4, P2 L31, P3 L31, P4 L27, P5 L2 "brackish water" instead of "freshened water"*
20    *P3 L24 "than during the rest of the year"*
*P4 L6 "one order of magnitude"*
*P5 L20 "data were obtained"*
*P6 L8 "than the difference"*
*P6 L23 "based on satellite data"*
25    *P6 L31 "based on CHL-a gradients"*
*P8 L17 "does not show"*
*P9 L3 "the relation"*
*P9 L7 "northeasterly"*
*P9 L8 "the relation"*
30    *P9 L16 "An A1 event causes …"*
*P9 L26: what do you mean? "are neither accompanied by … nor low surface salinity"*
*P9 L32 "northeasterly wind"*
*P12 L26 "In contrast, the volume …."*
*P12 L27 "wet periods" instead of "freshet periods"*

*P13 L1 "with a wind-forcing index T_v exceeding 500 km"*

*Fig. 3: impossible to see the circulation in the Azov Sea*

*Fig. 4: please connect the circles with a curve*

Thank you for many valuable grammatical and technical comments, they significantly helped to improve the quality of the text. We adopted your comments and remarks and made the corresponding corrections in the manuscript.

*Fig. 5: please provide correlation coefficients instead of the curves, i.e. delete Fig. 5*

Thank you for this comment. We deleted Fig. 5 and provided the related correlation coefficients in Section 4.1.

[revised manuscript text omitted]

---

## Author Response (AR3)

**Author's reply to comments from the Editor.**

Comments on Zavialov et al., supplementary material:

**Page 1:**

Delete the sentence "Because this model configuration is quite recent, …" Basin-averaged salinity Basin-averaged temperature but remains **rather** stable during the run The SSH also shows

**Page 2:**

2.Large-scale mean circulation The SSH and current patterns comma after (e.g. Staneva et al., 2011), (Figs. 2 and 3) does not appear
The mean SST (Fig. 1c) is in very good agreement ... deep Black Sea Figure 1, caption: ... (b) currents ...

**Page 3:**

(Figs. 2 and 3)

**Page 4:**

that does not **reflect** the instantaneous .... (no figure shown) ... consistent with the warming trend **in** the deep Black Sea ... The trend corresponds ... for **both the** Black Sea and Azov Sea **and amounts** to ~0.08°C/year ...

**Page 5:**

... **suggesting** that the warming trend ... for a day where of maximum in year 2003 (I do not understand the meaning) Perhaps: ... for a day when MLDs in 2003 were at its maxima ????

**Page 6:**

... in order **to** maintain ... Figure 6, caption: Instantaneous **c**urrents (top), **t**emperature (middle) and **s**alinity (bottom) ...

**Response:**

Thank you for your comments. The text was corrected.

**Supplementary Material to the paper untitled**

Water exchange between the Sea of Azov and the Black Sea through the Kerch Strait

by

Ivan Zavialov, Alexander Osadchiev, Roman Sedakov, Bernard Barnier, Jean-Marc Molines, and Vladimir Belokopytov

Assessment of the BSAS12 numerical model.

The BSAS12 model briefly described in the paper is a new configuration of the European ocean general circulation model NEMO for the Black Sea and the Azov Sea. BSAS12 is developed jointly at the Ocean Modelling Laboratory (OML) of the Shirshov Institute of Oceanology in Moscow and the MultiscalE Ocean Modelling (MEOM) group of the Institut des Géosciences de l'Environnement in Grenoble with the main objective to study the processes driving the exchanges between the Black Sea and the Azov Sea. Because this model configuration is quite recent, there is no reference for the moment. We present in this Supplementary Material a first assessment of the surface circulation produced the model simulation described in the paper.

*Figure 1: Time evolution of the basin averaged monthly mean Salinity (top), Temperature (middle), and SSH (bottom) for the whole duration of the simulation.*

**1. Model spin-up**

The model has been run from 1992 to 2017, and the time evolution of the basin-integrated quantities suggest two periods.

1992-2002: The basin-average Salinityaveraged salinity does not exhibit any trend during the first 10 years of the run (Fig. 1, top), indicating that evaporation, precipitation, river runoff and salt flux through the Bosphorus strait balance each other during this period. For that period, the basin-averaged Temperaturetemperature (Fig. 1, middle) shows a warming trend ( $0.025^{\circ}C/y$ ), an indication of a non-equilibrated heat balance. The Sea Surface Height (SSH, Fig. 1 bottom) shows a quick adjustment in the first year of the run (a drop of ~10 cm) but remains rather stable rather during the run.

2003-2017: After 2002, the basin-averaged salinity shows a small (~0.0012 psu/y) but regular positive trend. The basin-averaged temperature show a small decrease in the early 2000s but the warming trend resumes afterwards but is smaller. The SSH also showshows a small

drop after 2002. This suggest a change in the forcing fields (ERAinterim reanalysis) that needs to be investigated.

**2. Large-scale mean circulation**

The long-term time-mean large-scale circulation is shown in Fig. 12. The SSH and Currentcurrent patterns exhibit the major circulation patterns described in the literature: the Western and the Eastern cyclonic gyres and the Rim Current. The amplitude of the Rim Current (10 to 40 cm/s) is in good agreement with the geostrophic currents derived from altimetry and floats by Menna and Poulain (2015). The Sevastopol and the Batumi anticyclonic eddies, often reported in schematics of the circulation (e.g. Staneva et al., 2011), do not appear as strong features in the time-mean, although they are clearly among the most energetic features in the instantaneous flow (Fig.2 & Figs. 3 and 4). Their time and space variability is such that their signature in the mean does not appearsappear as a coherent eddy signal. The mean SST (Fig. 1e2c) is in very good agreement with the remote sensed estimates recently proposed for the period 1982-2015 by Sakalli and Basusta (2018), the modelled SST being slightly warmer. This difference could be explained by the different periods of averaging, the temperature increase over the deep blackBlack Sea being significant from the early 1990s (Sakkali et Batusta, 2018, Shapiro et al., 2010).

*Figure* 12: *Time mean for the period 1999-2009 of (a) SSH, (b)* *Currentscurrents* *and c) SST simulated by BSAS12 (the total duration of the run is 1992 to 2017).*

**3. Eddying circulation**

The horizontal grid-resolution of the model is ~6.75 km. We estimate the eddy-scale  $L_E$  to be  $L_E = \frac{1}{2}\lambda$ , with  $\lambda = 2\pi R_D$  being the length-scale associated to the first radius of deformation  $R_D$ . With this definition,  $L_E$  is an estimate of the characteristic eddy diameter. With  $R_D = ~20 \text{ km}$  in the open sea, an estimate of the eddy scale is  $L_E = ~60 \text{ km}$ . Therefore, there are 9 grid-points to resolve the eddy scale, which is enough to resolve the largest eddies. Nevertheless, we consider this model as eddy-permitting (and not eddy-resolving) because it does not properly resolve the shelf eddies that have a smaller characteristic lengthscale. We also mention that the use of the UBS advection scheme for momentum (Upstream-Biased Scheme, Shchepetkin and McWilliams, 2005, see also the NEMO documentation), which prevents the use of an explicit viscosity and significantly reduces the dissipation of the mesoscale instabilities. This allows the generation of numerous and energetics mesoscale eddies (Fig. 2&Figs. 3 and 4).

The instantaneous surface (5 m depth) circulation is illustrated in Figure 23 with two currents snapshots. At large scale, it agrees reasonably well with the previous analyses based on obervations or models (e.g. Staneva et al., 2001, Kovalev et al., 2003, Stanev, 2005, Menna and Poulain, 2015, Kubryakov et al., 2016, Miladinova et al., 2017). The main circulation features (i.e. the Rim Current, the Western and the Eastern cyclonic gyres, the Sevastopol and Batumi eddies) are well represented and exhibit realistic amplitudes (instantaneous currents between 30 to more than 60 cm/s) and a large variability.